# Homophily as a Lossy Channel: Decomposing Information in Graphs and Graph Neural Networks

**Vivek Kothari**
Department of Computer Science
University of Oxford
Oxford, UK
vivek.kothari@cs.ox.ac.uk

**Nicholas D. Lane**
Department of Computer Science and Technology
University of Cambridge
Cambridge, UK
ndl32@cam.ac.uk

## Abstract

Many GNN analyses reduce graph structure to label agreement (homophily), but in annotated graphs, edges often *gate* how neighbor information is interpreted. We model the directed edge as a computational channel relating a target label $X$, neighbor label $Z$, and edge annotation $Y$. We introduce the **Adjusted Information Profile** $\rho(P) = (\sigma^*, \rho_Y^*, \phi^*)$, which decomposes the information provided by the neighborhood into redundancy, unique information, and synergy. Crucially, we normalize these atoms by *structural channel capacities*, yielding bounded ratios comparable across diverse graphs. Leveraging a causal generative model that separates information emission from edge selection, we uncover the phenomenon of *Topological Masking*: we show that homophilous selection acts as a causal collider, driving observed synergy toward zero even when the underlying mechanism is highly synergistic. Across controlled synthetic sweeps and annotated benchmarks-including text-enriched citation and protein interaction networks-the profile accurately predicts the "Information Simplex" regime, distinguishing datasets where topological smoothing suffices from where edge-conditioned relational architectures are required.

## 1 Introduction and Background

Message-passing GNNs succeed when neighborhood aggregation preserves label-relevant signal Kipf & Welling (2017); Gilmer et al. (2017). Most dataset analyses reduce this condition to label agreement across edges (homophily) McPherson et al. (2001); Newman (2003a). This incomplete reduction motivates node-, edge-, and chance-corrected homophily metrics that aim to predict whether smoothing helps Zhu et al. (2020); Platonov et al. (2023a). It conflates two questions: what information is present in the neighborhood, and which pieces of the neighborhood are required for its computation.

In a graph Let $X$ denote the target label at the head of a directed edge, $Z$ the neighbor label at the tail, and $Y$ the edge annotation. A graph induces an edge-channel distribution $P(X, Z, Y)$. The central question becomes: how does $(Z, Y)$ inform $X$? Mutual information $I(X; (Z, Y))$ is insufficient because it collapses qualitatively different mechanisms into one scalar.

We instead decompose $I(X; (Z, Y))$ using Partial Information Decomposition (PID) Williams & Beer (2010). PID separates information into redundancy (agreement exploitable by smoothing), unique information (signal available from only one source), and synergy (signal that appears only from the joint observation of $Z$ and $Y$). Synergy captures edge-gated computation. We use the BROJA family, which defines redundancy via an optimization over couplings with fixed pairwise marginals Bertschinger et al. (2014).

Consider a citation graph where paper claims $X, Z \sim$ Bernoulli$(0.5)$ are independent, and citation semantics dictate $Y = X \oplus Z$. Standard homophily averages these edges to zero ($h = 0$), mischaracterizing the neighborhood as pure noise. Information-theoretically, while $Z$ or $Y$ individually yield $0$ bits of information about $X$, their combination yields exactly $1$ bit of synergistic information.

The Adjusted Information Profile (AIP) captures this deterministic relationship perfectly, yielding $\rho(P) = (1.0, 0.0, 0.0)$. This maximal synergy ($\sigma^* = 1.0$) mathematically guarantees that topological smoothing will fail (Topological Masking), proving that edge-gated architectures are strictly necessary. Full derivation is provided in Appendix A.3.

Naïve PID cannot be compared across datasets. Raw atoms scale with entropy and alphabet size, so cross-graph comparisons become ill-posed. Secondly, Homophilous edge selection induces selection bias: conditioning on an edge event acts as a collider and can suppress observable synergy even when the underlying mechanism is synergistic Pearl (2009).

We address both issues with the *Adjusted Information Profile* $\rho(P) = (\sigma^*, \rho_Y^*, \phi^*)$ (**Contribtion 1**). It normalizes PID atoms by matched structural channel capacities, yielding bounded ratios that remain comparable across graphs with different signal budgets. We also contribute a causal generative model that separates world emission from edge selection and exposes *Topological Masking* (**Contributions 2**). We then apply this to benchmark and new formulations of real world datasets (**Contribution 3**) showing that the information profile predicts when smoothing suffices versus when edge-conditioned relational computation is required.

## 2 ADJUSTED INFORMATION PROFILE (AIP)

We summarize the edge-channel information geometry by $\rho(P) = (\sigma^*, \rho_Y^*, \phi^*)$. Each component is a PID atom normalized by a matching structural capacity, so it is bounded and comparable across graphs. Appendix A gives design intuition and the more formal definition.

**Adjusted synergy.** $P_{\text{flat}}(x, z, y) = P(x, z)P(y \mid x)$. $C_\sigma := \sum_z P(z)\, C_{Y|X, Z=z} - C_{Y|X}$, where $C(\cdot)$ is Shannon capacity.

$$\sigma^*(P) := \frac{\text{Syn}(P) - \text{Syn}(P_{\text{flat}})}{C_\sigma - \text{Syn}(P_{\text{flat}})} \cdot \mathbf{1}\{C_\sigma > \text{Syn}(P_{\text{flat}})\}.$$

**Conditional unique resolution.** $P_\rightarrow(x, z, y) = P(x, z)P(y \mid z)$. Let $\text{Unq}_P(Y)$ abbreviate $\text{Unq}_P(X; Y \setminus Z)$ under BROJA.

$$\rho_Y^*(P) := \frac{\text{Unq}_P(Y) - \text{Unq}_{P_\rightarrow}(Y)}{H(X \mid Z) - \text{Unq}_{P_\rightarrow}(Y)} \cdot \mathbf{1}\{H(X \mid Z) > \text{Unq}_{P_\rightarrow}(Y)\}.$$

**Independence-weighted redundancy.** $Pind(x, z, y) = P(x)P(z \mid x)P(y \mid x)$.

$$\omega(P) := 1 - \frac{I(Z; Y \mid X)}{\min\{H(Z \mid X), H(Y \mid X)\}}.$$

Let $C_\phi$ denote the intersection capacity (Appendix B).

$$\phi^*(P) := \frac{\text{Red}(P) - \text{Red}(Pind)}{C_\phi - \text{Red}(Pind)} \cdot \mathbf{1}\{C_\phi > \text{Red}(Pind)\} \cdot \omega(P).$$

which downweights redundancy when sources are strongly coupled beyond $X$ (motivation in Appendix A, bounds in Appendix B).

*Remark* 2.1 (Scope of AIP and deterministic lifting). The AIP evaluates the observed edge-channel law $P_{\text{ec}}(X, Z, Y)$ under the strict assumption that the edge annotation $Y$ is a domain-native variable, not a label-derived surrogate. Deterministic lifting exposes the pitfalls of violating this assumption. If $Y = g(Z)$, the Markov chain $X \rightarrow Z \rightarrow Y$ yields $I(X; Z, Y) = I(X; Z), \quad I(X; Y \mid Z) = 0$, precluding $Y$ from contributing unique or synergistic information beyond $Z$. Conversely, if $Y = f(X, Z)$, $Y$ may trivially encode the target label $X$; any resulting nonzero PID atoms measure artifacts of the lifting rule rather than genuine exogenous edge semantics. Thus, label-derived lifting serves solely as an exploratory stress test, not as justification for applying AIP to unannotated graphs.

The AIP is well behaved and shows properties (boundedness, monotonicity under certain cases, etc) that make it more suitable for usage across details in dataset Appendix B. The main text uses the computable safe approximation as specified in the appendix.

## 3 THE CAUSAL PID–DCSBM

Annotated graphs are scarce; many benchmarks derive edge features from node labels, conflating topology and edge semantics. To study their joint influence under controlled conditions, we introduce a causal generator that explicitly separates *world emission* (the latent dyadic information) from *edge selection* (the observed graph). This factorization permits independent interventions on redundancy, unique information, and synergy (see Appendix C).

**World emission.** Node communities and sociabilities are drawn as $C_i \sim \text{Cat}(\pi)$ and $S_i \sim F_S$. For each ordered dyad $(i, j)$ we construct three sources of relational signal:

- **Redundancy** ($W_{ij}^{(R)}$): a Gaussian copula with correlation $\gamma_{\text{red}} \cdot \mathbf{1}\{C_i = C_j\}$.
- **Synergy** ($W_{ij}^{(S)}$): a Latin square $L(C_i, C_j)$ perturbed by label noise $\epsilon$; the synergy parameter is $\gamma_{\text{syn}} = 1 - 2\epsilon$.
- **Unique** ($W_{ij}^{(U)}$): a function $h(A_i, A_j)$ of attributes $A_i$ that are drawn independently of the communities $\mathbf{C}$.

Full templates and parametrisation details are provided in Appendix C.

**Edge selection.** Edges are sampled from a degree-corrected logistic model:

$$\Pr(E_{ij} = 1) = \sigma\Big(\beta_0 + \log S_i S_j + \beta_C \tilde{U}_{C,ij} + \beta_R \tilde{W}_{ij}^{(R)} + \beta_S \tilde{W}_{ij}^{(S)} + \beta_U \tilde{W}_{ij}^{(U)}\Big), \quad (1)$$

where $\tilde{U}_{C,ij} = \mathbf{1}\{C_i = C_j\} - \mu_C$ is centered, and the dyadic components $\tilde{W}^{(\cdot)}$ are centred and scaled. The intercept $\beta_0$ is calibrated to achieve a target edge density $\delta$ (see Appendix C).

**Topological Masking.** Homophilous edge selection acts as a collider: conditioning on the edge event biases the dyad law away from the world distribution $P_{\text{pp}}$ to the observed $P_{\text{ec}}$. Even when the world mechanism is strongly synergistic, this selection can suppress the synergy visible in $P_{\text{ec}}$.

**Theorem 3.1** (Topological Masking (informal))**.** *Let the world mechanism be synergistic, i.e.* $\text{Syn}(P_{\text{pp}}) > 0$. *Under increasingly homophilous selection (large $\beta_C$ in equation 10), the edge-conditioned law $P_{\text{ec}}$ concentrates on dyads with $X = Z$, and the observed adjusted synergy vanishes:* $\sigma^*(P_{\text{ec}}) \to 0$.

A formal statement, the proof, and an inverse-probability-weighted (IPW) recovery of $P_{\text{pp}}$ under positivity are deferred to Appendix C.9. The IPW procedure enables identification of the world information profile even when the observed graph is masked by homophilous selection.

## 4 EXPERIMENTS

Our evaluation targets three mechanistic claims. First, $\rho(P)$ recovers the intended PID atoms under controlled interventions in the causal PID–DCSBM. Second, homophilous selection can suppress *observed* synergy under $P_{\text{ec}}$ while leaving *world* synergy under $P_{\text{pp}}$ high (Topological Masking). Third, architectural gains from edge-conditioned models increase with $\sigma^*$ and decrease with $\phi^*$. Appendix E reports all numerics, seeds, confidence intervals, capacity routines, and hyperparameter ranges. Appendix D formalizes why homophily cannot certify edge-gated computation.

**Claim 1: Controlled traversal of the information simplex.** We instantiate the PID–DCSBM generator and sweep one mechanism at a time while matching density and marginal entropies up to estimation error. The profile follows the intended ordering across regimes. Synergy-targeted sweeps push $\sigma^*$ toward one while keeping $\rho_Y^*$ and $\phi^*$ small. Redundancy-targeted sweeps increase $\phi^*$ and contract $\sigma^*$. Exact sweep grids, alphabet sizes, and recovery curves appear in Appendix E

**Claim 2: Topological Masking under homophilous selection.** We fix a synergistic world emission law and increase the selection strength that biases edges toward $X = Z$. Selection increases $I(X; Z)$ and reduces $H(X \mid Z)$ in the edge-conditioned law. As a result, $\sigma^*(P_{\text{ec}})$ collapses toward zero even when $\sigma^*(P_{\text{pp}})$ remains high. Appendix E

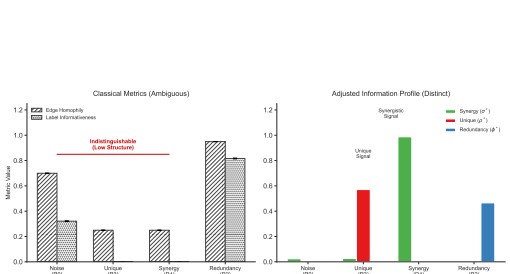

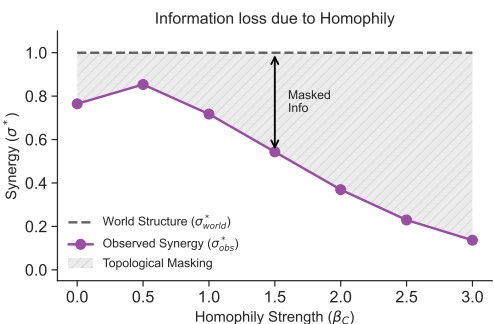

Figure 1: **Metric Discrimination.** Comparison of Classical Metrics (left) versus the Adjusted Information Profile (right). Classical metrics fail to distinguish between Unique (R3) and Synergy (R4) scenarios, rendering them indistinguishable. In contrast, the adjusted metrics ($\sigma^*, \rho^*, \phi^*$) successfully disambiguate the scenarios, correctly identifying the Unique and Synergistic signals.

Figure 2: **Topological Masking.** As homophily pressure $\beta_C$ increases, the observable synergy $\sigma_{ec}$ (solid line) diverges from the latent ground truth synergy $\sigma_{pp}$ (dashed line). The shaded region represents the information gap caused by the topological suppression of edge-dependent signals.

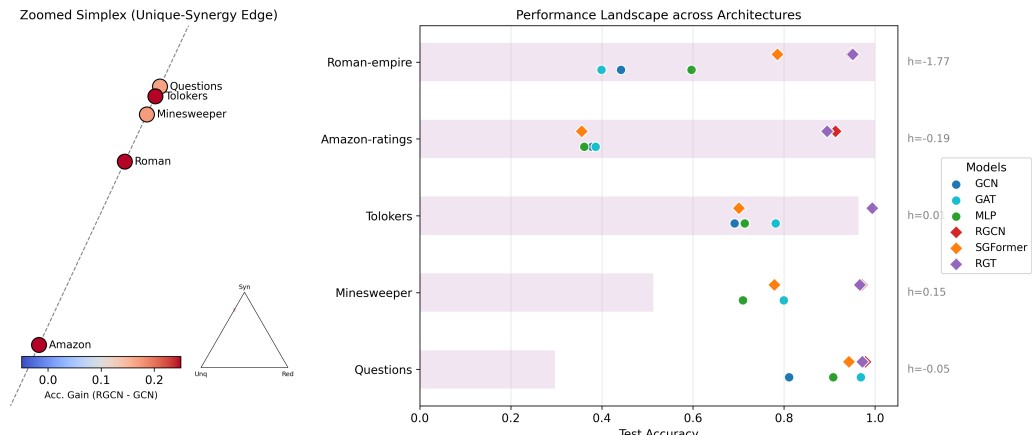

Figure 3: **The Information Simplex predicts GNN failure modes. (Left)** Datasets projected onto the Unique-Synergy edge of the Adjusted Information Profile simplex. The color gradient represents the performance gap ($\Delta$) between relational (RGCN) and smoothing (GCN) architectures; red indicates a large gap. **(Right)** Performance landscape sorted by Synergy ($\sigma^*$). High synergy (top) necessitates relational models (Diamonds), leading to a significant divergence from smoothing baselines (Circles). Conversely, feature-dominant datasets (bottom) show negligible architectural variance despite high synergy. Classical homophily ($h$, right axis) fails to capture this structural ordering.

**Claim 3: Architectural gains align with simplex position.** We compare smoothing architectures (GCN, GAT) against edge-conditioned models (RGCN and transformer variants). Across synthetic sweeps, the edge-conditioned advantage grows with $\sigma^*$ and shrinks with $\phi^*$, consistent with synergy capturing the need for edge-gated composition rather than label agreement. Appendix E reports additional details.

**Real graphs.** We report two settings. (i) *Unannotated heterophily benchmarks:* For standard heterophily benchmarks without native edge annotations, AIP is not directly identifiable as an edge-semantic diagnostic. We therefore report label-derived latent lifting only as an exploratory stress test: it asks whether a hypothetical coarse edge variable could separate regimes that homophily alone conflates. Because such lifted variables are deterministic functions of node labels, they may either add no information beyond $Z$ or leak information about $X$ by construction (Remark 2.1). Accordingly, our primary empirical claims rely on domain-native annotated graphs. (ii) *Annotated benchmarks:* we create (truely domain based edge annotative graphs) and evaluate a text-enriched citation graph (TEG) and a multi-relational protein interaction graph (MR-PPI). In both cases, the simplex position

matches the observed gap between smoothing and edge-conditioned models. Dataset construction and validation checks appear in Appendix E

## 5 CONCLUSION

We introduced a capacity-normalized PID diagnostic for annotated graphs. The profile separates redundancy, unique annotation signal, and synergy in a way homophily cannot. A causal selection model reveals Topological Masking: homophilous selection can eliminate observable synergy without changing the world mechanism. Across controlled sweeps and annotated benchmarks, the profile predicts when smoothing should work and when edge-conditioned computation is necessary.

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

# A ADJUSTED INFORMATION PROFILE: MOTIVATION, NULL MODELS, AND CAPACITIES

This appendix motivates the Adjusted Information Profile $\rho(P) = (\sigma^*, \rho_Y^*, \phi^*)$ and formalizes the defintions. We focus on three design decisions: (i) modeling annotated graphs by the edge-channel law $P(X, Z, Y)$, (ii) using explicit null models to isolate mechanisms, and (iii) normalizing PID atoms by matched structural capacities rather than by total mutual information or entropy. Appendix B states formal properties and Appendix C.5 specifies estimation details.

**Edge-channel setup.** Each directed edge $(j \rightarrow i)$ yields a triple $(X, Z, Y)$, where $X$ is the target label at node $i$, $Z$ is the neighbor label at node $j$, and $Y$ is the edge annotation. The empirical edge-conditional distribution $P_{ec}(X, Z, Y)$ defines the channel family induced by the dataset. We write $P$ when the distinction is not needed.

**PID atoms.** We decompose the total information that $(Z, Y)$ provides about $X$ as

$$I(X; Z, Y) = \text{Red}(X; Z, Y) + \text{Unq}(X; Z \setminus Y) + \text{Unq}(X; Y \setminus Z) + \text{Syn}(X; Z, Y),$$

using the BROJA family Bertschinger et al. (2014) throughout our experiments.

## A.1 WHY HOMOPHILY IS NOT A COMPUTATIONAL DESCRIPTOR

Homophily depends only on $(X, Z)$ through agreement or mixing statistics. It ignores the edge annotation $Y$. This omission matters whenever $Y$ changes how a neighbor should be interpreted. A citation edge tagged `extends` versus `criticizes` can reverse the predictive meaning of a neighboring label. Similarly, an inhibitory protein interaction conveys different evidence than an activating interaction even when endpoint labels match. In these cases, the same level of label agreement can correspond to different computations. The profile targets this gap by measuring how $Y$ gates the informativeness of $Z$ about $X$.

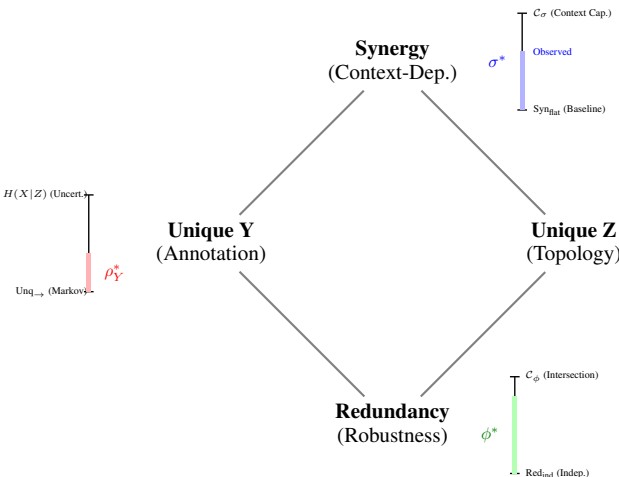

Figure 4: **The Adjusted Information Profile on the PID Lattice.** Standard PID atoms (nodes) are normalized not by total information, but by their specific structural capacities (bars). $\sigma^*$ measures how much of the available contextual bandwidth is utilized; $\rho_Y^*$ measures annotation innovation relative to topological uncertainty; $\phi^*$ measures robustness relative to intersection capacity.

## A.2 WHY MUTUAL INFORMATION IS NOT ENOUGH

$I(X; Z, Y)$ aggregates all dependence between $X$ and the neighborhood variables $(Z, Y)$. It does not identify the mechanism by which information arises. PID separates three modes that correspond to distinct algorithmic biases: (i) redundancy supports smoothing because multiple sources convey overlapping evidence for $X$, (ii) unique information in $Y$ supports relation-specific shortcuts, and (iii) synergy supports edge-gated composition because $Z$ becomes informative only after conditioning on $Y$. The XOR example makes this distinction concrete. For noiseless XOR, $I(X; Z) = I(X; Y) = 0$ but $I(X; Z, Y) > 0$, so the information is purely synergistic.

## A.3 A TOY EXAMPLE

To build intuition for the Adjusted Information Profile (AIP) and the phenomenon of Topological Masking, we formally expand the XOR Citation Graph introduced in Example 1.

**1. The Generative Process.** Let the target node label $X \in \{0, 1\}$ and the neighbor node label $Z \in \{0, 1\}$ be independent and uniformly distributed, such that $P(X = x, Z = z) = 0.25$ for all $x, z$. We define the directed edge annotation $Y \in \{0, 1\}$ as the deterministic XOR function of the incident nodes: $Y = X \oplus Z$.

**2. The Failure of Homophily.** Standard homophily simply measures the alignment between $X$ and $Z$. Because $X$ and $Z$ are independent, $P(X = Z) = 0.5$. Once adjusted for chance, the global homophily score is $h = 0$. A standard GCN interpreting this graph sees only random noise in its neighborhood and will learn a weight matrix approaching zero to ignore the neighbors, bounded entirely by the node's self-features.

**3. The Nature of the Information (Raw PID Atoms).** We assess the mutual information provided by the neighborhood. Individually, neither the neighbor's label $Z$ nor the edge annotation $Y$ provides any information about $X$:

$$I(X; Z) = 0 \text{ bits}, \quad I(X; Y) = 0 \text{ bits} \tag{2}$$

The edge acts as a cryptographic key: without the edge $Y$, the neighbor $Z$ is useless. Without the neighbor $Z$, the edge $Y$ is useless. However, the joint mutual information is perfectly resolving:

$$I(X; Z, Y) = H(X) - H(X|Z, Y) = 1 - 0 = 1 \text{ bit} \tag{3}$$

**1. Flat Bundle ($P_{\text{flat}}$)**     **2. Markov ($P_\rightarrow$)**     **3. Independent ($P_{\text{ind}}$)**

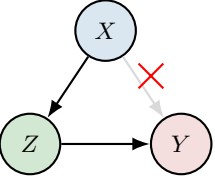
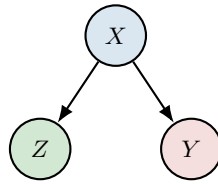

Removes context-
dependence.
Baseline for $\sigma^*$.

Forces $X \rightarrow Z \rightarrow Y$.
Baseline for $\rho_Y^*$.

Sources indepen-
dent given $X$.
Baseline for $\phi^*$.

Figure 5: **Causal Ablations defining the baselines.** The three adjusted metrics rely on specific null models ($P_{\text{flat}}, P_\rightarrow, P_{\text{ind}}$) constructed by surgically severing specific dependencies (red crosses) while preserving marginals. This isolates the mechanism of interest (synergy, unique innovation, or redundancy) from background correlations.

Applying the BROJA PID framework, the raw information atoms decompose this exactly:

$$\text{Redundancy } (R) = 0 \text{ bits} \tag{4}$$
$$\text{Unique } Z(U_Z) = 0 \text{ bits} \tag{5}$$
$$\text{Unique } Y(U_Y) = 0 \text{ bits} \tag{6}$$
$$\text{Synergy } (S) = 1 \text{ bit} \tag{7}$$

**4. The Adjusted Information Profile (AIP).** While the raw PID atoms give us the absolute information in bits, they are unbound and cannot be compared across datasets with different class sizes or entropies. We map these atoms to the AIP bounded simplex $\rho(P) = (\sigma^*, \rho_Y^*, \phi^*)$. Because the synergistic capacity of this deterministic channel is 1 bit, and the raw synergy is 1 bit, the normalization yields:

$$\rho(P) = \left(\frac{1}{1}, \frac{0}{1}, \frac{0}{1}\right) = (1.0, 0.0, 0.0) \tag{8}$$

**5. Architectural Implications.** This extreme profile coordinate isolates the root cause of the structural disparity observed between simple and edge-conditioned GNNs. Because the profile is concentrated entirely in the Synergy dimension ($\sigma^* = 1.0$), architectures that aggregate $Z$ independently of $Y$ (like GCN or GAT) will permanently destroy the 1 bit of available information during the smoothing phase. The graph strictly requires a relational, edge-gated message-passing mechanism (like an RGCN or Graph Transformer) that computes a joint function of $(Z, Y)$ before aggregating.

## A.4  Why we normalize by structural capacities

Raw PID atoms scale with entropy and alphabet size. This scaling breaks cross-graph comparability. A dataset with a larger label alphabet can exhibit larger Syn solely because $H(X)$ is larger. Normalizing by total mutual information $I(X; Z, Y)$ also fails in regimes where $I(X; Z, Y)$ is small or unstable, and it collapses the distinction between "more information exists" and "the mechanism is more synergistic." We instead normalize each atom by a matched structural capacity that upper bounds how large that atom can be under the channel constraints induced by the dataset. The resulting ratios are bounded and interpretable as fractions of the available structural bandwidth.

## A.5  Null models as mechanism isolators

Each ratio is defined by contrasting the observed law $P$ against a null model that removes a specific dependency while preserving others. These nulls serve two roles. First, they isolate mechanisms by construction. Second, they provide a baseline for capacity normalization.

**Flat-bundle null (removes context dependence).** We define

$$P_{\text{flat}}(x, z, y) := P(x, z) P(y \mid x).$$

Equivalently, $P_{\text{flat}}(y \mid x, z) = P(y \mid x)$. This preserves the pairwise relations $(X, Z)$ and $(X, Y)$ but removes any additional dependence of $Y$ on $Z$ beyond $X$. Under $P_{\text{flat}}$, $Y$ cannot act as a context gate. We therefore interpret $\text{Syn}(P) - \text{Syn}(P_{\text{flat}})$ as synergy attributable to edge-conditioned gating rather than to marginal association.

**Markov-degraded null (forces $X \to Z \to Y$).** We define

$$P_\to(x, z, y) := P(x, z) \, P(y \mid z),$$

so $P_\to(y \mid x, z) = P(y \mid z)$. This enforces the degraded structure in which edge annotations do not carry information about $X$ beyond what is already present in $Z$. The difference $\text{Unq}_P(X; Y \setminus Z) - \text{Unq}_{P_\to}(X; Y \setminus Z)$ isolates edge-specific signal that is not explainable by neighbor labels alone.

**Conditional-independence null (enforces $Z \perp Y \mid X$).** We define

$$P_{\text{ind}}(x, z, y) := P(x) \, P(z \mid x) \, P(y \mid x).$$

This preserves the $(X, Z)$ and $(X, Y)$ marginals and removes residual coupling between sources beyond the target. It provides a baseline redundancy level expected when overlap is mediated only through $X$.

A.6  CHANNEL CAPACITIES: WHAT THEY MEASURE AND WHY THEY MATCH THE RATIOS

A capacity is an operational ceiling. It answers: how much information can this channel convey about $X$ under optimal input usage? Capacities allow comparisons across graphs by normalizing against a dataset-specific ceiling induced by the channel structure. We use three capacities, matched to the three ratios.

**Capacity for adjusted synergy.** Synergy in our setting measures additional information about $X$ that becomes available when $Y$ is interpreted in the context of $Z$. The flat-bundle null removes this context. We therefore normalize by a *contextual capacity* that measures the maximal gain in channel capacity when conditioning on $Z$:

$$\mathcal{C}_\sigma := \sum_z P(z) \, C_{Y|X, Z=z} \ - \ C_{Y|X},$$

where $C_{Y|X, Z=z}$ is the Shannon capacity of the conditional channel $P(y \mid x, z)$ at fixed $z$ and $C_{Y|X}$ is the capacity of the marginal channel $P(y \mid x)$. If $P(y \mid x, z) = P(y \mid x)$ for all $z$, then $C_{Y|X, Z=z} = C_{Y|X}$ and $\mathcal{C}_\sigma = 0$. In that case, the dataset provides no structural bandwidth for edge-gated interpretation, and $\sigma^*$ is defined to be zero by convention. we estimate capacities with Blahut-Arimoto AlgorithmBlahut (1972).

**Capacity for unique edge resolution.** Unique information in $Y$ about $X$ cannot exceed the residual uncertainty in $X$ after observing $Z$. This yields the natural ceiling $H(X \mid Z)$. We subtract the Markov baseline because unique-$Y$ can be nonzero under finite-sample noise even when $X \to Z \to Y$ approximately holds. This yields a ratio that measures how much of the remaining label uncertainty $Y$ resolves beyond what $Z$ already provides.

**Capacity for redundancy.** Redundancy measures information about $X$ that both $Z$ and $Y$ can convey. The maximal overlap is constrained by how well each source channel separates labels. We normalize by an *intersection capacity* $\mathcal{C}_\phi$ that upper bounds extractable overlap given the two conditional channels $P(z \mid x)$ and $P(y \mid x)$. The full definition and its relation to pairwise KL separations appear in Appendix B. Here we note the operational interpretation: $\mathcal{C}_\phi$ is large when both channels discriminate labels similarly, and it is small when their discriminative directions are orthogonal.

A.7  ADJUSTED RATIOS AND CALIBRATION UNDER NULLS

We restate the ratios for completeness.

**Adjusted synergy.**

$$\sigma^*(P) := \frac{\mathrm{Syn}(P) - \mathrm{Syn}(P_{\mathrm{flat}})}{\mathcal{C}_\sigma - \mathrm{Syn}(P_{\mathrm{flat}})} \cdot \mathbb{I}\{\mathcal{C}_\sigma > \mathrm{Syn}(P_{\mathrm{flat}})\}.$$

If $Y$ does not depend on $Z$ given $X$, then $P = P_{\mathrm{flat}}$ and $\sigma^*(P) = 0$. If edge annotations are strongly context-gated, then $\mathrm{Syn}(P)$ approaches the contextual ceiling and $\sigma^*(P)$ increases toward one.

**Conditional unique resolution.**   Let $\mathrm{Unq}_P(Y)$ denote $\mathrm{Unq}_P(X; Y \setminus Z)$. Then

$$\rho_Y^*(P) := \frac{\mathrm{Unq}_P(Y) - \mathrm{Unq}_{P_\rightarrow}(Y)}{H(X \mid Z) - \mathrm{Unq}_{P_\rightarrow}(Y)} \cdot \mathbb{I}\{H(X \mid Z) > \mathrm{Unq}_{P_\rightarrow}(Y)\}.$$

If the Markov condition $X \rightarrow Z \rightarrow Y$ holds, then $P = P_\rightarrow$ and $\rho_Y^*(P) = 0$.

**Independence-weighted redundancy.**   Let $\mathrm{Red}(P)$ abbreviate $\mathrm{Red}_P(X; Z, Y)$. We define the independence weight

$$\omega(P) := 1 - \frac{I(Z; Y \mid X)}{\min\{H(Z \mid X), H(Y \mid X)\}},$$

which lies in $[0, 1]$ (Appendix B) and downweights redundancy when sources are strongly coupled beyond $X$. Then

$$\phi^*(P) := \frac{\mathrm{Red}(P) - \mathrm{Red}(P_{\mathrm{ind}})}{\mathcal{C}_\phi - \mathrm{Red}(P_{\mathrm{ind}})} \cdot \mathbb{I}\{\mathcal{C}_\phi > \mathrm{Red}(P_{\mathrm{ind}})\} \cdot \omega(P).$$

This ratio calibrates to zero under conditional independence beyond the target, and it approaches one when overlap saturates the intersection ceiling while sources remain weakly coupled given $X$.

**Practical computation.**   We discretize continuous variables to finite alphabets (default $B{=}8$ bins for continuous features), apply additive smoothing ($\delta{=}10^{-6}$), compute BROJA PID by convex optimization, and report bootstrap uncertainty (1000 resamples).

Boundedness, calibration under the ablations, consistency, and stability proofs appear in Appendix B.

## A.8   WHY CAPACITIES COMPLEMENT NULL-MODEL NORMALIZATION

Null models can also calibrate information decomposition by comparing to an ensemble that preserves coarse statistics. NuMIT proposes a null-based approach for comparing decompositions across complex systems Liardi et al. (2025). Our goal differs. We normalize by capacities tied to the edge-channel family induced by annotated graphs and selection. This choice makes the ratios comparable across graphs even when a meaningful null ensemble is difficult to define. The two perspectives are compatible. NuMIT-style nulls can act as significance tests for whether a measured ratio exceeds what is expected under randomized edge semantics, while capacity ratios quantify how close the measured dependence is to the structural ceiling.

## A.9   TOPOLOGICAL MASKING AND WHY THE PROFILE MAKES IT VISIBLE

Topological masking is a selection effect. Homophilous selection increases agreement between $X$ and $Z$ under the observed edge law $P_{\mathrm{ec}}$. This reduces $H(X \mid Z)$ and constrains the range over which synergy can contribute. Consequently, $\sigma^*(P_{\mathrm{ec}})$ can be small even when the world emission law is highly synergistic. Appendix C.8 formalizes this mechanism and gives an IPW identification result.

## A.10   ARCHITECTURAL INTERPRETATION

The three ratios correspond to three inductive biases. Large $\phi^*$ indicates redundant neighborhood evidence and favors smoothing. Large $\rho_Y^*$ indicates that edge labels provide direct resolution of $X$ and favors models that consume edge features. Large $\sigma^*$ indicates that $Y$ gates the interpretation of $Z$ and favors edge-conditioned composition, as in typed message passing or relational attention. The experiments in Appendix E show that the observed performance gap between smoothing and relational models tracks these regimes.

# B    THEORETICAL FOUNDATIONS

## B.1    ASSUMPTIONS

Let $\mathcal{X}, \mathcal{Z}, \mathcal{Y}$ be finite alphabets. Let $P$ denote the true distribution on $\mathcal{X} \times \mathcal{Z} \times \mathcal{Y}$ and $P_n$ the empirical distribution from $n$ i.i.d. edge samples.

**(A1) Interior support.** $\exists \alpha > 0$: $P(x, z, y) \geq \alpha$ for all $(x, z, y)$, or estimator uses additive smoothing.

**(A2) PID regularity.** The PID functional is continuous and locally Lipschitz on the simplex interior.

**(A3) Nondegenerate denominators.** $\exists \kappa > 0$: $\mathcal{C}_\sigma - \mathrm{Syn}_{\mathrm{flat}} \geq \kappa$, $H(X|Z) - \mathrm{Unq}_\rightarrow(Y) \geq \kappa$, $\mathcal{C}_\phi - \mathrm{Red}_{\mathrm{ind}} \geq \kappa$.

**(A4) Identification region.** $\mathrm{Syn}_{\mathrm{flat}} \leq \mathrm{Syn}_P \leq \mathcal{C}_\sigma$, $\mathrm{Unq}_\rightarrow(Y) \leq \mathrm{Unq}_P(Y) \leq H(X|Z)$, $\mathrm{Red}_{\mathrm{ind}} \leq \mathrm{Red}_P \leq \mathcal{C}_\phi$.

## B.2    PROOFS

**Boundedness.** Under (A4), each ratio has form $(\mathrm{atom} - \mathrm{baseline})/(\mathrm{capacity} - \mathrm{baseline})$ with numerator and denominator in $[0, \mathrm{capacity} - \mathrm{baseline}]$. The indicator enforces nonzero denominator. For $\phi^*$, multiplication by $\omega(P) \in [0, 1]$ preserves $[0, 1]$.

**Consistency.** $\|P_n - P\|_1 = O_p(n^{-1/2})$ by standard results. Each component of $\boldsymbol{\rho}$ is locally Lipschitz under (A1)–(A3): entropies by Fannes–Audenaert, KL by smoothing, PID atoms by (A2), capacity proxies as suprema of Lipschitz functionals. Denominators bounded below by $\kappa$. Hence $\|\boldsymbol{\rho}(P_n) - \boldsymbol{\rho}(P)\| = O_p(n^{-1/2})$.

**Stability.** Changing one edge changes $P_n$ by at most $2/n$ in $\ell_1$. For $k$ edits, $\|P_n(G) - P_n(G')\|_1 \leq 2k/n$. Apply Lipschitz bound.

**Calibration.** (i) $Y \perp Z|X$ implies $P = P_{\mathrm{flat}}$, so $\mathrm{Syn}_P = \mathrm{Syn}_{\mathrm{flat}}$ and $\sigma^* = 0$. (ii) $X \rightarrow Z \rightarrow Y$ implies $P = P_\rightarrow$, so $\mathrm{Unq}_P(Y) = \mathrm{Unq}_\rightarrow(Y)$ and $\rho_Y^* = 0$. (iii) $Y = f(Z)|X$ implies $I(Z; Y|X) = H(Y|X)$, so $\omega(P) = 0$ and $\phi^* = 0$.

## B.3    ASYMPTOTIC NORMALITY

**Proposition B.1.** *Under (A1)–(A3) and Fréchet differentiability of $\boldsymbol{\rho}$ at $P$:*
$$\sqrt{n}(\widehat{\boldsymbol{\rho}}_n - \boldsymbol{\rho}(P)) \Rightarrow \mathcal{N}(0, \Sigma(P)).$$

*Proof.* Multinomial CLT gives $\sqrt{n}(P_n - P) \Rightarrow \mathcal{N}(0, \Sigma_0)$. Apply functional delta method.    □

## B.4    NULL MODEL CONSTRUCTIONS

**Flat-bundle null.** $P_{\mathrm{flat}}(x, z, y) := P(x, z)P(y|x)$.

**Markov-degraded null.** $P_\rightarrow(x, z, y) := P(x, z)P(y|z)$.

**Independence null.** $P_{\mathrm{ind}}(x, z, y) := P(x)P(z|x)P(y|x)$, enforcing $Z \perp Y|X$.

## B.5    CAPACITY COMPUTATION

$\mathcal{C}_\sigma$. For each $z$, compute $\max_Q I_Q(X; Y|Z = z)$ via Blahut–Arimoto Blahut (1972). Compute $\max_Q I_Q(X; Y)$ for the marginal channel. Take difference weighted by $P(z)$.

$\mathcal{C}_\phi$. Apply add-$\delta$ smoothing ($\delta = 10^{-6}$) to avoid infinite KL. Optimize over $Q(X)$ via projected gradient descent.

## B.6 CAUSAL IDENTIFICATION

Under the SCM in Section 3 and positivity ($p_{ij} \geq \underline{p} > 0$), the world law is identified by IPW:

$$\mathbb{E}_{P_{\mathrm{pp}}}[h(X, Z, Y)] = \frac{\mathbb{E}_{P_{\mathrm{ec}}}[h(X, Z, Y) \cdot p_{ij}^{-1}]}{\mathbb{E}_{P_{\mathrm{ec}}}[p_{ij}^{-1}]}.$$

The Hájek estimator with estimated propensities $\hat{p}_{ij}$ recovers $P_{\mathrm{pp}}$ from selected edges.

## B.7 ADDITIONAL PROPERTIES

We establish several additional properties of the adjusted profile.

### B.7.1 PERMUTATION INVARIANCE

**Proposition B.2** (Label invariance). *Let $\pi_X, \pi_Z, \pi_Y$ be bijections on $\mathcal{X}, \mathcal{Z}, \mathcal{Y}$ respectively. Define $P^\pi(x, z, y) := P(\pi_X^{-1}(x), \pi_Z^{-1}(z), \pi_Y^{-1}(y))$. Then $\boldsymbol{\rho}(P^\pi) = \boldsymbol{\rho}(P)$.*

*Proof.* Entropy, mutual information, KL divergence, and PID atoms are invariant under symbol relabeling. The ablations and capacities inherit this invariance. □

### B.7.2 MARKOV SIMPLIFICATION

**Proposition B.3** (Vanishing baseline for $\rho_Y^*$). *Under the Markov-degraded ablation,* $\mathrm{Unq}_{P_\to}(Y) = 0$. *Hence $\rho_Y^* = \mathrm{Unq}_P(Y)/H(X|Z)$ when $H(X|Z) > 0$.*

*Proof.* Under $P_\to(x, z, y) = P(x, z)P(y|z)$, we have $X \perp Y \mid Z$, so $I_{P_\to}(X; Y|Z) = 0$. For BROJA, $\mathrm{Unq}(Y) = \min_{Q \in \Delta} I_Q(X; Y|Z)$ where $\Delta$ constrains marginals. Since $P_\to \in \Delta$ achieves zero, $\mathrm{Unq}_{P_\to}(Y) = 0$. □

### B.7.3 NON-NEGATIVITY OF CONTEXTUAL CAPACITY

**Proposition B.4** (Contextual capacity is non-negative). *$\mathcal{C}_\sigma \geq 0$, with equality iff $P(Y|X, Z = z) = P(Y|X)$ for all $z$ (no context dependence).*

*Proof.* Let $C(W)$ denote the capacity of channel $W$. The marginal channel is $P_{Y|X} = \sum_z P(z)P_{Y|X, Z=z}$. By concavity of capacity in the channel:

$$C\Big(\sum_z P(z)P_{Y|X,Z=z}\Big) \leq \sum_z P(z)C(P_{Y|X,Z=z}).$$

Rearranging gives $\mathcal{C}_\sigma \geq 0$. Equality holds iff the concavity bound is tight, which occurs iff all conditional channels are identical. □

### B.7.4 INDEPENDENCE WEIGHT BOUNDS

**Proposition B.5** (Weight is well-defined). *$\omega(P) \in [0, 1]$ for all distributions $P$.*

*Proof.* By properties of conditional mutual information: $I(Z; Y|X) \leq H(Z|X)$ and $I(Z; Y|X) \leq H(Y|X)$. Hence $I(Z; Y|X) \leq \min\{H(Z|X), H(Y|X)\}$, giving $\omega \geq 0$. Upper bound $\omega \leq 1$ is immediate from non-negativity of mutual information. □

### B.7.5 SYMMETRIC UNIQUE RESOLUTION

**Definition B.1** (Symmetric profile). Define $\rho_Z^*(P)$ analogously to $\rho_Y^*$, measuring unique information from topology $Z$ beyond annotation $Y$:

$$\rho_Z^*(P) := \frac{\mathrm{Unq}_P(Z) - \mathrm{Unq}_\leftarrow(Z)}{H(X|Y) - \mathrm{Unq}_\leftarrow(Z)} \cdot \mathbb{I}\{H(X|Y) > \mathrm{Unq}_\leftarrow(Z)\},$$

where $P_\leftarrow(z|x, y) := P(z|y)$ enforces $X \to Y \to Z$.

**Proposition B.6** (Duality). *$\rho_Y^* = 1$ and $\rho_Z^* = 0$ iff $Y$ is a sufficient statistic of $(Z, Y)$ for $X$.*

### B.7.6 DATA PROCESSING

**Proposition B.7** (Coarsening reduces resolution). *Let $\tilde{Y} = g(Y)$ for some deterministic function g. Then $\rho^*_{\tilde{Y}}(P) \leq \rho^*_Y(P)$.*

*Proof.* By data processing, $\mathrm{Unq}(\tilde{Y}) \leq \mathrm{Unq}(Y)$. The ceiling $H(X|Z)$ is unchanged. Hence the ratio decreases. $\square$

### B.7.7 TENSORIZATION

**Proposition B.8** (Product distributions). *Let $(X_1, Z_1, Y_1) \perp (X_2, Z_2, Y_2)$ with product distribution $P_\otimes$. For $X = (X_1, X_2)$, $Z = (Z_1, Z_2)$, $Y = (Y_1, Y_2)$:*

$$\mathrm{Syn}_{P_\otimes} = \mathrm{Syn}_{P_1} + \mathrm{Syn}_{P_2}, \quad \mathrm{Unq}_{P_\otimes}(Y) = \mathrm{Unq}_{P_1}(Y_1) + \mathrm{Unq}_{P_2}(Y_2).$$

*The adjusted ratios do not tensorize simply due to capacity normalization.*

*Proof.* PID atoms are additive under independence Williams & Beer (2010). Capacities generally do not decompose additively. $\square$

### B.7.8 CONCENTRATION INEQUALITIES

**Proposition B.9** (Hoeffding-type bound). *Under (A1)–(A3), for any $t > 0$:*

$$\mathbb{P}\big(|\hat{\sigma}^*_n - \sigma^*| > t\big) \leq 2\exp\Big(-\frac{n\kappa^2 t^2}{2L_\sigma^2}\Big),$$

*where $L_\sigma$ is the Lipschitz constant of $\mathrm{Syn}(\cdot)$ and $\kappa$ bounds denominators away from zero. Analogous bounds hold for $\rho^*_Y$ and $\phi^*$.*

*Proof.* Apply McDiarmid's inequality. Changing one observation changes $P_n$ by $O(1/n)$ in total variation, hence changes Lipschitz functionals by $O(L/n)$. The bounded differences condition gives the exponential tail. $\square$

### B.7.9 MONOTONICITY IN CHANNEL QUALITY

**Proposition B.10** (Degradation ordering). *If channel $P'(Y|X, Z)$ is a degraded version of $P(Y|X, Z)$ (i.e., $P' = P \circ W$ for some stochastic matrix W), then:*

$$\sigma^*(P') \leq \sigma^*(P), \quad \rho^*_Y(P') \leq \rho^*_Y(P).$$

*Proof.* Channel degradation reduces capacity and unique information by data processing. Baselines computed under degraded channels also decrease, but the ratio decreases overall. $\square$

### B.7.10 RELATIONSHIP TO INTERACTION INFORMATION

**Proposition B.11** (Connection to co-information). *Let $\mathrm{CoI}(X; Y; Z) = I(X; Y) - I(X; Y|Z)$ denote co-information. Then:*

$$\mathrm{Syn}_P - \mathrm{Red}_P = -\mathrm{CoI}(X; Y; Z).$$

*When $\mathrm{CoI} < 0$ (synergy-dominant), $\mathrm{Syn}_P > \mathrm{Red}_P$.*

*Proof.* This follows from the PID decomposition and properties of co-information Williams & Beer (2010). $\square$

## B.7.11 CONVEXITY OF THE IDENTIFICATION REGION

**Proposition B.12** (Convex domain). *The set of distributions $P$ satisfying (A4) is convex in the probability simplex.*

*Proof.* Each constraint in (A4) is a linear inequality in $P$ (entropies and PID atoms are concave/convex, and the bounds involve fixed nulls computed from $P$). The intersection of half-spaces is convex. □

*Remark.* The profile $\boldsymbol{\rho}(P)$ itself is not convex in $P$ due to the ratio structure, but continuity ensures well-behaved variation over convex combinations.

## B.7.12 IDENTIFICATION REGION CHARACTERIZATION

We provide conditions under which (A4) holds.

**Proposition B.13** (Sufficient conditions for (A4)). *Assumption (A4) holds if:*

*(a) $P$ has full support on $\mathcal{X} \times \mathcal{Z} \times \mathcal{Y}$;*

*(b) The channels $P(Y|X, Z = z)$ are not all identical (for $\sigma^*$);*

*(c) $Y$ is not a deterministic function of $Z$ (for $\rho_Y^*$);*

*(d) $Z$ and $Y$ are not deterministically related given $X$ (for $\phi^*$).*

*Proof.* (a) ensures all entropies and capacities are well-defined and positive. (b) ensures $\mathcal{C}_\sigma > 0$. (c) ensures $H(X|Z) > 0$ or $\mathrm{Unq}(Y) < H(X|Z)$. (d) ensures $\omega(P) > 0$. □

**Proposition B.14** (Boundary behavior). *At the boundary of (A4):*

- *If $\mathcal{C}_\sigma = \mathrm{Syn}_{\text{flat}}$: all channels identical, $\sigma^* = 0$ by definition.*
- *If $H(X|Z) = \mathrm{Unq}_{\rightarrow}(Y) = 0$: $X$ determined by $Z$ and no unique info from $Y$, $\rho_Y^* = 0$.*
- *If $\mathcal{C}_\phi = \mathrm{Red}_{\text{ind}}$: no redundant distinguishing power, $\phi^* = 0$.*

*The indicators in the definitions ensure graceful degradation at boundaries.*

## B.7.13 SCALE INVARIANCE

**Proposition B.15** (Invariance under alphabet extension). *Adding a zero-probability symbol to any of $\mathcal{X}, \mathcal{Z}, \mathcal{Y}$ does not change the profile.*

*Proof.* Entropy, mutual information, and PID atoms depend only on positive-probability outcomes. Zero-probability extensions do not affect these quantities. □

## B.7.14 RELATIONSHIP TO CHANNEL CAPACITY

**Proposition B.16** (Capacity interpretation). *Let $C_{Y|X}$ denote the capacity of channel $P(Y|X)$ and $C_{Y|X,Z}$ the conditional capacity $\sum_z P(z)C(P(Y|X, Z = z))$. Then:*

$$\mathcal{C}_\sigma = C_{Y|X,Z} - C_{Y|X}.$$

*This is the* state information *in the language of channels with states Cover & Thomas (2006).*

**Proposition B.17** (Operational meaning). *$\sigma^*$ measures the fraction of state information utilized: if a genie reveals $Z$, what fraction of the additional channel capacity is reflected in the synergy structure of $P$?*

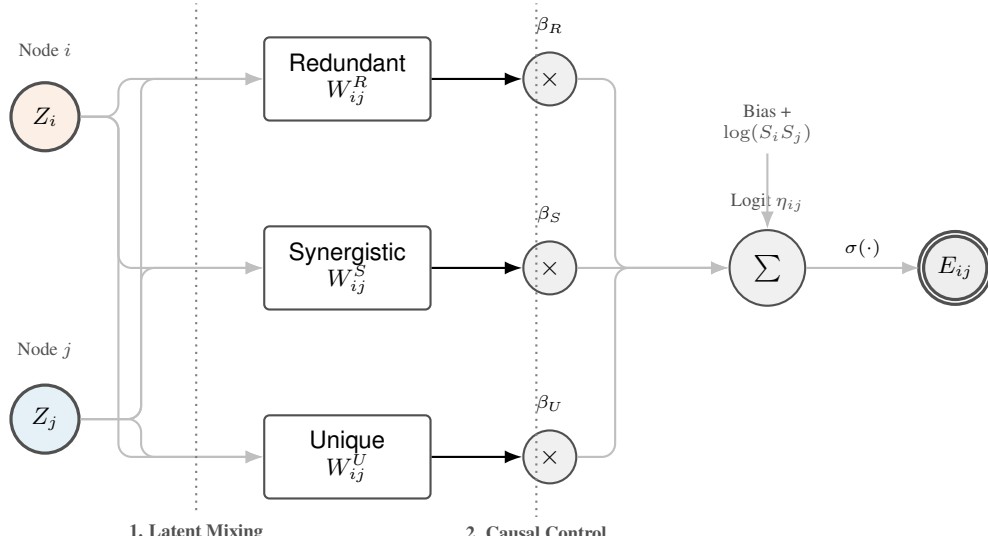

Figure 6: **Causal architecture of the PID-DCSBM (Schematic).** The generative process is visualized as a feed-forward circuit. Latent variables from node pairs $(Z_i, Z_j)$ are routed into three isolated channels. The contributions are gated by $\beta$ coefficients—the causal intervention points—before aggregation. This design ensures that redundancy, synergy, and unique information are structural properties of the graph generation, not post-hoc artifacts.

## C    PID–DCSBM: COMPLETE SPECIFICATION AND CAUSAL SEMANTICS

This appendix gives the full definition of the PID–DCSBM used in our synthetic experiments. The model serves two purposes. First, it generates annotated graphs whose edge-channel law $P(X, Z, Y)$ occupies controlled regions of the Information Simplex. Second, it separates *world emission* from *edge selection*, which lets us make causal statements about how topology can mask edge-gated computation.

**Observed edge-channel variables.**    For a directed edge $(j \rightarrow i)$, we define $X := C_i$ (target label), $Z := C_j$ (neighbor label), and $Y := W_{ij}$ (edge annotation). We treat $C_i$ as the node label for synthetic prediction tasks; this keeps the masking analysis explicit. All results extend to the case where $X$ is a noisy function of $C_i$, but we do not use that variant in the main experiments.

### C.1    STRUCTURAL CAUSAL MODEL AND INTERVENTIONS

The generator is a structural causal model with node-level latents, dyad-level emissions, and dyad-level selection. We write $\sigma(t) = 1/(1 + e^{-t})$ for the logistic function.

**Node-level latents.**    For each node $i \in [n]$:

$$C_i \sim \text{Cat}(\pi) \in [K], \qquad S_i \sim F_S,$$

where $C_i$ is the class label and $S_i$ is a sociability factor (degree correction).

**Dyad-level world emission.**    For each ordered dyad $(i, j)$ with $i \neq j$, we emit three dyad components

$$W_{ij} = \left( W_{ij}^{(R)}, W_{ij}^{(S)}, W_{ij}^{(U)} \right),$$

using the channels in Section C.4. The emission knobs $(\gamma_{\text{red}}, \gamma_{\text{syn}}, \gamma_{\text{unq}})$ control the strength of redundancy-, synergy-, and unique-dominant mechanisms in the world law.

**Edge selection (observed topology).**    Conditioned on $(\mathbf{C}, \mathbf{S}, \mathbf{W})$, edges are sampled independently as

$$\mathbb{P}(E_{ij} = 1 \mid \mathbf{C}, \mathbf{S}, \mathbf{W}) = \sigma(\eta_{ij}), \tag{9}$$

with logit score

$$\eta_{ij} = \beta_0 + \log(S_i S_j) + \beta_C \, \tilde{U}_{C,ij} + \beta_R \, \tilde{W}_{ij}^{(R)} + \beta_S \, \tilde{W}_{ij}^{(S)} + \beta_U \, \tilde{W}_{ij}^{(U)}. \tag{10}$$

Here $\tilde{U}_{C,ij} = \mathbf{1}\{C_i = C_j\} - \mu_C$ is the centered homophily indicator and $\tilde{W}_{ij}^{(\cdot)}$ denotes a centered (and optionally standardized) version of the corresponding dyad component. The coefficients $(\beta_C, \beta_R, \beta_S, \beta_U)$ isolate causal effects of selection mechanisms.

**Causal statements (what the generator identifies).** Because the model separates emission from selection, we can interpret parameter changes as interventions. In particular:

- $\mathrm{do}(\beta_C \leftarrow b)$ changes selection toward or away from homophily while holding the world emission $P_{\mathrm{pp}}(X, Z, Y)$ fixed.
- $\mathrm{do}(\gamma_{\mathrm{syn}} \leftarrow g)$ changes the world emission mechanism that produces synergistic edge annotations, affecting both $P_{\mathrm{pp}}$ and the observed edge-conditioned law $P_{\mathrm{ec}}$.
- Comparing $P_{\mathrm{pp}}$ to $P_{\mathrm{ec}}$ isolates selection bias. This is the operational content of Topological Masking.

We use these interventions to generate monotone sweeps in the main experiments and to attribute changes in $\sigma^*$ to selection rather than emission.

### C.2 World and edge-conditioned laws

The generator induces two distributions over $(X, Z, Y)$.

**Definition C.1** (World and edge-conditioned laws). The *world law* is the dyad distribution over all ordered pairs

$$P_{\mathrm{pp}}(x, z, y) = \mathbb{P}(C_i = x, \, C_j = z, \, W_{ij} = y), \quad i \neq j.$$

The *edge-conditioned law* is the observed distribution given that an edge exists:

$$P_{\mathrm{ec}}(x, z, y) = \mathbb{P}(C_i = x, \, C_j = z, \, W_{ij} = y \mid E_{ij} = 1).$$

**Lemma C.1** (Selection reweights the world law). *Let $p(x, z, y) := \mathbb{P}(E_{ij} = 1 \mid C_i = x, C_j = z, W_{ij} = y)$ denote the propensity under equation 9–equation 10. Then*

$$P_{\mathrm{ec}}(x, z, y) = \frac{P_{\mathrm{pp}}(x, z, y) \, p(x, z, y)}{\mathbb{E}_{P_{\mathrm{pp}}}[p(X, Z, Y)]}.$$

*Proof.* By Bayes, $P_{\mathrm{ec}}(x, z, y) \propto P_{\mathrm{pp}}(x, z, y)\mathbb{P}(E_{ij} = 1 \mid x, z, y)$. Normalizing by $\mathbb{P}(E_{ij} = 1) = \mathbb{E}_{P_{\mathrm{pp}}}[p(X, Z, Y)]$ yields the identity. $\qquad\square$

Lemma C.1 is the key link between causal selection and the information profile. It implies that changing selection coefficients alters $P_{\mathrm{ec}}$ even when $P_{\mathrm{pp}}$ is unchanged.

### C.3 Algorithm (directed and undirected variants)

Algorithm 1 states the directed version, which matches the edge-channel definition $(j \rightarrow i)$ used throughout. For undirected graphs, we either (i) generate ordered dyads and symmetrize by keeping an undirected edge if $E_{ij} = 1$ or $E_{ji} = 1$, or (ii) generate only $i < j$ and copy $W_{ij}$ to both directions. We use option (ii) in the synthetic experiments unless stated otherwise.

### C.4 Emission channels

The emission channels control the world information structure. Each channel is designed to affect a specific PID mode of $P_{\mathrm{pp}}(X, Z, Y)$ while keeping marginals stable enough for controlled sweeps. In all cases, we discretize emitted values to a finite alphabet $\mathcal{Y}$ before computing PID .

---

**Algorithm 1** PID–DCSBM Generation (directed)

---

**Require:** $n, K, \pi, F_S$; emission knobs $(\gamma_{\mathrm{red}}, \gamma_{\mathrm{syn}}, \gamma_{\mathrm{unq}})$; selection coefficients $(\beta_C, \beta_R, \beta_S, \beta_U)$; target density $\delta$; RNG seed $s$.

**Ensure:** Directed graph $G$ and dyad attributes $\{W_{ij}\}_{i \neq j}$.

 1: Set RNG seed $s$.
 2: Sample $C_i \sim \mathrm{Cat}(\pi)$ and $S_i \sim F_S$ for all $i \in [n]$.
 3: **for** each ordered dyad $(i, j)$ with $i \neq j$ **do**
 4:     Emit $W_{ij} = (W_{ij}^{(R)}, W_{ij}^{(S)}, W_{ij}^{(U)})$ using Section C.4.
 5: **end for**
 6: Compute centering constants $\mu_C = \mathbb{E}[\mathbf{1}\{C_i = C_j\}]$ and $\mu_{W^{(\cdot)}} = \mathbb{E}[W_{ij}^{(\cdot)}]$ under the world law (Monte Carlo estimate).
 7: Calibrate $\beta_0$ so that $\mathbb{E}[\sigma(\eta_{ij})] = \delta$ (Section C.5).
 8: **for** each ordered dyad $(i, j)$ with $i \neq j$ **do**
 9:     Form centered covariates $\tilde{U}_{C,ij}$ and $\tilde{W}_{ij}^{(\cdot)}$.
10:     Compute $\eta_{ij}$ by equation 10 and sample $E_{ij} \sim \mathrm{Bern}(\sigma(\eta_{ij}))$.
11: **end for**
12: Return $G = ([n], \{(i, j) : E_{ij} = 1\})$ and $\{W_{ij}\}$.

---

**Synergy channel (Latin-square / group-difference template).** Let $L : [K] \times [K] \to [K]$ be a Latin square. We use the group-difference Latin square

$$L(a, b) := (a - b) \bmod K,$$

which reduces to XOR when $K = 2$ and has a constant diagonal $L(a, a) = 0$ for all $a$. This choice makes the masking analysis transparent. We then define a symmetric noise model with noise rate $\epsilon_{\mathrm{syn}} \in [0, 1]$:

$$W_{ij}^{(S)} = \begin{cases} L(C_i, C_j) & \text{with probability } 1 - \epsilon_{\mathrm{syn}}, \\ U, \ U \sim \mathrm{Unif}([K]) & \text{with probability } \epsilon_{\mathrm{syn}}. \end{cases}$$

We parameterize noise by $\gamma_{\mathrm{syn}} := 1 - \epsilon_{\mathrm{syn}}$, so larger $\gamma_{\mathrm{syn}}$ yields stronger synergy.

**Redundancy channel (agreement-correlated edge semantics).** Redundancy in our edge-channel setting reflects overlap between neighbor evidence ($Z = C_j$) and edge semantics ($Y$). We induce this overlap by generating an edge annotation that correlates with label agreement. Let $A_{ij} := \mathbf{1}\{C_i = C_j\}$. We generate a continuous score $T_{ij}$ whose distribution is fixed and whose separation across $A_{ij} \in \{0, 1\}$ is controlled by $\gamma_{\mathrm{red}}$:

$$T_{ij} \mid A_{ij} = a \sim \mathcal{N}((2a - 1)m, \ 1), \qquad m = m(\gamma_{\mathrm{red}}) \geq 0,$$

and we set

$$W_{ij}^{(R)} := Q_W(\Phi(T_{ij})),$$

where $\Phi$ is the standard normal CDF and $Q_W$ is the quantile function of a target marginal on $\mathcal{Y}$. This is a Gaussian-copula construction: it fixes the marginal of $W_{ij}^{(R)}$ while tuning dependence on agreement through $m(\gamma_{\mathrm{red}})$. In practice we use $m(\gamma_{\mathrm{red}}) = \mathrm{arctanh}(\gamma_{\mathrm{red}})$ for $\gamma_{\mathrm{red}} \in [0, 0.95]$.

**Unique channel (head-label encoding templates).** Unique information in $Y$ about $X = C_i$ should not be explainable by $Z = C_j$. We implement this by encoding the head label in $Y$ independent of the tail label. We provide two templates.

*Template U1 (noisy injective code).* Fix an injective map $M : [K] \to \mathcal{Y}$. With noise rate $\epsilon_{\mathrm{unq}} := 1 - \gamma_{\mathrm{unq}}$,

$$W_{ij}^{(U)} = \begin{cases} M(C_i) & \text{with probability } 1 - \epsilon_{\mathrm{unq}}, \\ U, \ U \sim \mathrm{Unif}(\mathcal{Y}) & \text{with probability } \epsilon_{\mathrm{unq}}. \end{cases}$$

This makes $Y$ resolve $X$ even when $Z$ is weakly informative.

*Template U2 (optional neighbor-conditioned encoding).* For each $z \in [K]$, let $M_z : [K] \to \mathcal{Y}$ be injective. Define

$$W_{ij}^{(U)} = M_{C_j}(C_i) + \text{noise}.$$

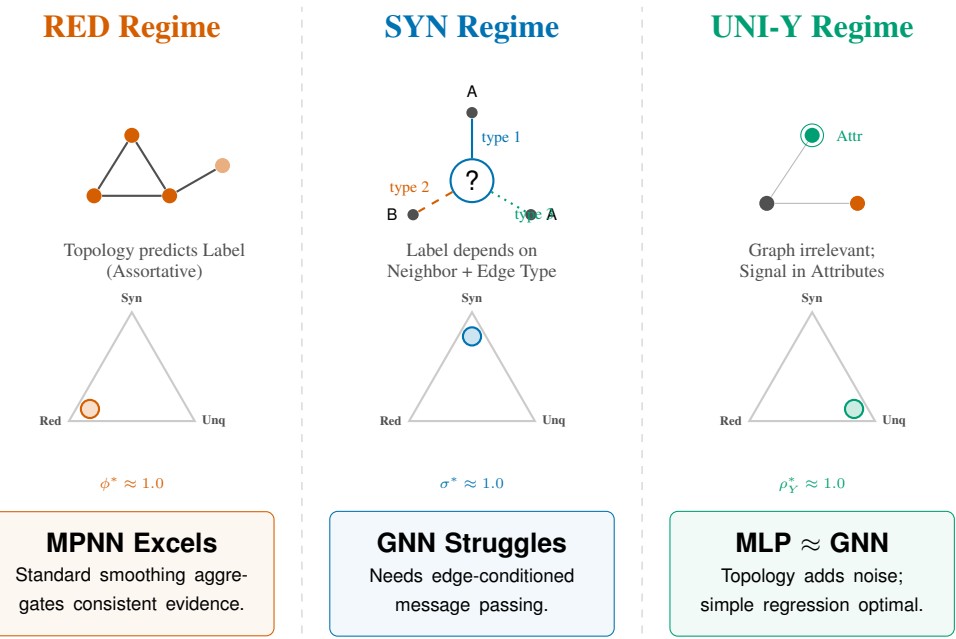

Figure 7: **Atlas of Experimental Regimens.** A comparison of the three primary data-generating regimes in the PID-DCSBM. **Top Row:** The relationship between topology and labels. Note how Synergy (Center) requires distinct edge types (dashed/dotted) to decode the signal. **Middle Row:** The target informational footprint on the PID simplex. **Bottom Row:** The downstream implication for Graph Neural Network architecture choice.

If $M_z$ is the same map for all $z$, this reduces to U1 and is purely unique. If $M_z$ varies strongly with $z$, the channel becomes context-gated and can introduce synergy. We use this template only for controlled ablations reported in Appendix E; the main synthetic regimens use U1 to avoid mixing unique and synergistic effects.

**Assembling the annotation alphabet.** In the simplest regimens, we set the observed edge annotation to a single component: $Y = W^{(R)}$, $Y = W^{(S)}$, or $Y = W^{(U)}$. When we need a mixed regime, we define $Y$ as the tuple $(W^{(R)}, W^{(S)}, W^{(U)})$ and then discretize or hash it into $\mathcal{Y}$. Appendix E reports the exact choice per regimen.

## C.5 DENSITY CALIBRATION

We calibrate $\beta_0$ so that the expected edge density equals a target $\delta$. Define

$$D(\beta_0) = \mathbb{E}\big[\sigma(\eta_{ij}(\beta_0))\big],$$

where the expectation is over the world law of $(C_i, C_j, S_i, S_j, W_{ij})$.

**Lemma C.2** (Existence and uniqueness). *$D(\beta_0)$ is continuous and strictly increasing, with $D(-\infty) = 0$ and $D(+\infty) = 1$. Hence for any $\delta \in (0,1)$ there exists a unique $\beta_0^\star$ such that $D(\beta_0^\star) = \delta$.*

*Proof.* $\sigma(\cdot)$ is continuous and strictly increasing, and $\eta_{ij}$ is affine in $\beta_0$. Monotone convergence yields the limits as $\beta_0 \to \pm\infty$. Strict monotonicity follows because $\sigma(\eta_{ij}(\beta_0))$ is strictly increasing in $\beta_0$ almost surely. $\square$

**Implementation.** We approximate $D(\beta_0)$ by Monte Carlo over a subsample of ordered dyads and solve by bisection to tolerance $10^{-4}$. We report the number of iterations and runtime in Appendix E.

Table 1 lists the default regimens used in the main paper. All regimens use $K = 4$, $\pi = (0.25, 0.25, 0.25, 0.25)$, $F_S = \text{Gamma}(2, 2)$ normalized to mean 1, and target density $\delta = 0.01$. Appendix E reports the sweep grids and the number of graph realizations per point.

Table 1: PID–DCSBM regimens used in the main experiments.

| Regimen | $\gamma_{\text{red}}$ | $\gamma_{\text{syn}}$ | $\beta_C$ | $\beta_R$ | $\beta_S$ | $\beta_U$ |
|---|---|---|---|---|---|---|
| R0 (Noise) | 0 | 0 | 0.0 | 0.0 | 0.0 | 0.0 |
| R1 (Selection) | 0 | 0 | 2.0 | 0.0 | 0.0 | 0.0 |
| R2 (RED) | 0.8 | 0 | 1.0 | 1.0 | 0.0 | 0.0 |
| R3 (UNI) | 0 | 0 | 0.5 | 0.0 | 0.0 | 1.0 |
| R4 (SYN) | 0 | 0.9 | 0.5 | 0.0 | 1.0 | 0.0 |
| R5 (MASK) | 0 | 0.9 | 3.0 | 0.0 | 0.5 | 0.0 |

## C.7   STRUCTURAL PROPERTIES

**Theorem C.1** (Node exchangeability). *For any permutation $\tau$ of $[n]$, the relabeled graph $\tau(G)$ has the same distribution as $G$.*

*Proof.* The node latents $(C_i, S_i)$ are i.i.d. across nodes. Dyad emissions depend on $(C_i, C_j)$ only through equivariant functions, and selection depends on $(C_i, C_j, S_i, S_j, W_{ij})$ through equation 10, which is equivariant under relabeling. $\qquad\square$

**Proposition C.1** (Conditional independence of edges). *Conditional on $(\mathbf{C}, \mathbf{S}, \mathbf{W})$, the edge indicators $\{E_{ij}\}_{i \neq j}$ are mutually independent.*

*Proof.* Given $(\mathbf{C}, \mathbf{S}, \mathbf{W})$, each $\eta_{ij}$ is deterministic and edges are sampled independently by equation 9. $\qquad\square$

## C.8   TOPOLOGICAL MASKING: FORMAL ANALYSIS

We formalize masking as the causal effect of homophilous selection on the edge-conditioned synergy. Throughout, we work with the group-difference Latin square, so $W_{ij}^{(S)} = 0$ whenever $C_i = C_j$ in the noiseless case.

**Proposition C.2** (Masking under homophilous selection). *Consider the noiseless synergy channel $(\gamma_{\text{syn}} = 1)$ with $W_{ij}^{(S)} = L(C_i, C_j)$ and $L(a, a) = 0$ for all $a$. If $\beta_C > 0$ and the selection rule equation 10 increases $\mathbb{P}(C_i = C_j \mid E_{ij} = 1)$ relative to the world law, then*

$$\text{Syn}(P_{\text{ec}}) < \text{Syn}(P_{\text{pp}}).$$

*Proof.* Let $q_{\text{ec}} = \mathbb{P}(C_i = C_j \mid E_{ij} = 1)$ and $q_{\text{pp}} = \mathbb{P}(C_i = C_j)$. Under $C_i = C_j$, the noiseless output satisfies $W_{ij}^{(S)} = 0$ deterministically, so the restricted law has zero synergy. Under $C_i \neq C_j$, the mapping $L(C_i, C_j)$ is bijective in each argument, so the restricted law preserves the pure-synergy structure. Since $P_{\text{ec}}$ is a mixture of these two restrictions with mixture weight $q_{\text{ec}} > q_{\text{pp}}$, the synergy under $P_{\text{ec}}$ is strictly smaller than under $P_{\text{pp}}$. $\qquad\square$

**Corollary C.1** (Masking magnitude bound). *Let $\Delta_{\text{Syn}} := \text{Syn}(P_{\text{ec}}) - \text{Syn}(P_{\text{pp}})$. Under the conditions of Proposition C.2,*

$$\Delta_{\text{Syn}} \leq -\text{Syn}(P_{\text{pp}} \mid C_i \neq C_j)\,(q_{\text{ec}} - q_{\text{pp}}).$$

**Proposition C.3** (No masking under selection independent of $(X, Z, Y)$). *If $\beta_C = \beta_R = \beta_S = \beta_U = 0$ and $S_i \equiv 1$, then $p(x, z, y)$ is constant and $P_{\text{ec}} = P_{\text{pp}}$. Consequently, $\Delta_{\text{Syn}} = 0$.*

*Proof.* If $p(x, z, y)$ is constant, Lemma C.1 yields $P_{\text{ec}} = P_{\text{pp}}$. $\qquad\square$

## C.9 IPW CORRECTION AND CAUSAL IDENTIFICATION

Proposition C.4 states the identification result used in the main paper. It follows directly from Lemma C.1.

**Proposition C.4** (IPW identification of the world law). *Assume* positivity*: there exists $\underline{p} > 0$ such that $p(x, z, y) \geq \underline{p}$ on the support of $P_{\mathrm{pp}}$. Then*

$$P_{\mathrm{pp}}(x, z, y) = \frac{P_{\mathrm{ec}}(x, z, y)\, p(x, z, y)^{-1}}{\mathbb{E}_{P_{\mathrm{ec}}}[p(X, Z, Y)^{-1}]}.$$

*Proof.* Rearrange the identity in Lemma C.1 and normalize. $\square$

**Plug-in estimation of world-law PID.** PID atoms are functionals of a distribution. Proposition C.4 identifies $P_{\mathrm{pp}}$, so we estimate $P_{\mathrm{pp}}$ by IPW and then apply the same PID estimator used for $P_{\mathrm{ec}}$. We report an IPW stress test under known propensities in Appendix E.

## C.10 SYNERGY CHANNEL PROPERTIES

**Proposition C.5** (Pure synergy under a Latin square). *If $(C_i, C_j)$ is uniform on $[K]^2$ and $W_{ij}^{(S)} = L(C_i, C_j)$ for a Latin square $L$, then*

$$I(C_i; W^{(S)}) = I(C_j; W^{(S)}) = 0, \qquad I(C_i, C_j; W^{(S)}) = \log K.$$

*Under BROJA, the information is allocated to synergy and* $\mathrm{Syn} = \log K$.

*Proof.* For fixed $c_i$, $L(c_i, \cdot)$ is a permutation, so $W^{(S)}$ is uniform and $I(C_i; W^{(S)}) = 0$. The same holds for $C_j$. Since $W^{(S)}$ is a deterministic function of $(C_i, C_j)$ with uniform range, $H(W^{(S)}) = \log K$ and $I(C_i, C_j; W^{(S)}) = \log K$. The BROJA coupling constraints preserve the zero pairwise informations, so the mass is allocated to synergy. $\square$

## C.11 REDUNDANCY CHANNEL PROPERTIES

The redundancy template is designed to increase the overlap between $Z = C_j$ and $Y = W^{(R)}$ about $X = C_i$ by correlating $Y$ with agreement $A_{ij}$. A complete characterization of PID redundancy under discretization depends on the chosen binning and PID definition. We record a conservative monotonic proxy.

**Proposition C.6** (Mutual information proxy increases with separation). *In the redundancy channel, the mutual information $I(A_{ij}; W_{ij}^{(R)})$ is non-decreasing in $m(\gamma_{\mathrm{red}})$, and hence non-decreasing in $\gamma_{\mathrm{red}}$ for monotone $m(\cdot)$.*

*Proof.* $W_{ij}^{(R)}$ is a monotone transform of $T_{ij}$. The family $\{T_{ij} \mid A_{ij} = a\}$ forms a binary Gaussian location model whose separation is $2m$. Larger $m$ increases the Bayes optimal classification accuracy of $A_{ij}$ from $T_{ij}$, which implies larger $I(A_{ij}; T_{ij})$. Data processing yields $I(A_{ij}; W_{ij}^{(R)}) \geq 0$ and preserves monotonicity under the quantile transform. $\square$

**Empirical monotonicity of redundancy.** In the synthetic sweeps, $\phi^*$ increases monotonically with $\gamma_{\mathrm{red}}$ under fixed selection (Appendix E). We treat this empirical monotonicity as the relevant guarantee for the generator's intended use.

## D HOMOPHILY METRICS AND THEIR FAILURE MODES ON ANNOTATED GRAPHS

This appendix formalizes common homophily metrics and explains why they cannot characterize edge-gated computation. We also analyze their behavior under homophilous selection, which drives Topological Masking.

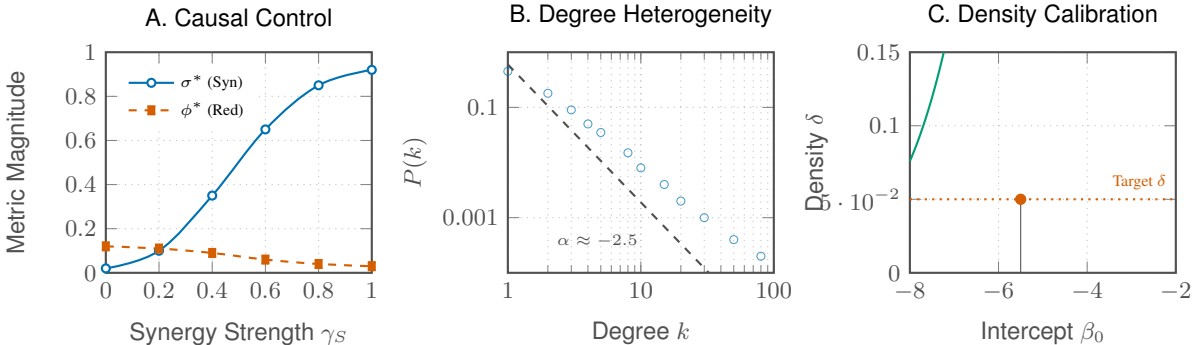

Figure 8: **Validation of PID-DCSBM properties.** **(A)** Tuning the synergy channel parameter $\gamma_S$ produces a clean, monotonic response in the Adjusted Synergy metric $\sigma^*$, confirming precise causal control. **(B)** The generative process maintains realistic heavy-tailed degree distributions (slope $\approx -2.5$) essential for GNN benchmarking. **(C)** The expected edge density $D(\beta_0)$ is strictly monotonic, enabling stable and efficient calibration to sparse targets.

## D.1 CLASSICAL DEFINITIONS

Let $G = (V, E)$ be a graph with node labels $X_i \in \{1, \ldots, K\}$. Let $(i, j) \in E$ denote a directed edge.

- **Edge homophily** (or **graph homophily**) is the empirical fraction of edges that connect nodes with the same label:

$$h_{\text{edge}} = \frac{1}{|E|} \sum_{(i,j) \in E} \mathbf{1}\{X_i = X_j\}. \tag{11}$$

- **Node homophily** averages homophily over nodes, which can be more informative under class imbalance or skewed degree distributions. For node $i$, its *node homophily* is the fraction of its neighbors sharing its label. The graph's node homophily is the average over all labeled nodes:

$$h_{\text{node}} = \frac{1}{|V_L|} \sum_{i \in V_L} \frac{|\{j \in \mathcal{N}(i) : X_j = X_i\}|}{|\mathcal{N}(i)|}, \tag{12}$$

where $V_L \subseteq V$ is the set of labeled nodes and $\mathcal{N}(i)$ denotes the neighbors of $i$.

- **Adjusted (edge) homophily** corrects for chance agreement due to class proportions Newman (2003b):

$$h_{\text{adj}} = \frac{h_{\text{edge}} - \sum_{c=1}^{K} \pi_c^2}{1 - \sum_{c=1}^{K} \pi_c^2}, \tag{13}$$

where $\pi_c = \mathbb{P}(X = c)$ is the class prior estimated from the labeled nodes.

- **Label informativeness** (LI) Platonov et al. (2023b) measures how much knowing the label of one node reduces uncertainty about the label of a random neighbor. It is defined as the mutual information between the labels of adjacent nodes, normalized by the entropy of the neighbor's label:

$$\text{LI} = \frac{I(X_i; X_j)}{H(X_j)}, \quad \text{for } (i, j) \in E. \tag{14}$$

LI ranges from $0$ (labels independent) to $1$ (neighbor label perfectly predictable). Unlike homophily, LI can capture asymmetric predictive relationships and is invariant to label permutations.

Platonov et al.Platonov et al. (2023b) discuss the instability of $h_{\text{edge}}$ and $h_{\text{adj}}$ under common heterophily benchmarks and advocate for label informativeness as a more robust structural descriptor.

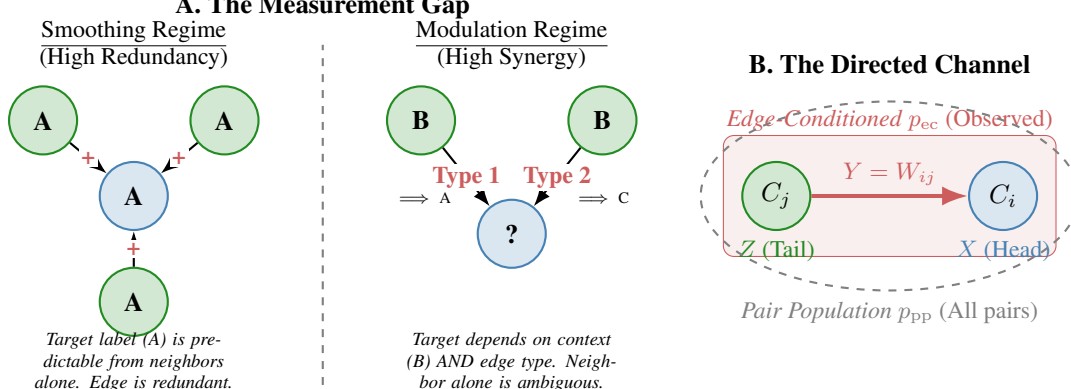

Figure 9: **The failure of node-centric homophily.** (A) In smoothing regimes, labels propagate simply. In modulation regimes, the edge annotation $Y$ acts as a gate or switch for the context $Z$, creating high synergy that label agreement metrics miss. (B) The directed-edge view defines the variables $(X, Z, Y)$ and distinguishes the latent population of all pairs ($p_{\text{pp}}$) from the observed distribution ($p_{\text{ec}}$), exposing selection effects.

Closest to our work is $LI$. While it measures the total signal in the neighbor label, it cannot determine if that signal is redundant with the edge annotation $Y$ or if it requires $Y$ as a contextual gate (synergy). Our profile isolates these through the independence-weighted redundancy $\phi^*$ and the contextual capacity normalization

### D.2 ANNOTATED GRAPHS INDUCE A DIFFERENT OBJECT

Our paper studies triples $(X, Z, Y)$ per directed edge. $X$ is the target label at the destination node. $Z$ is the neighbor label at the source node. $Y$ is the edge annotation. Homophily depends only on $(X, Z)$ through $\mathbf{1}\{X = Z\}$. It ignores $Y$. This is not a cosmetic omission. In edge-gated tasks, the meaning of $Z$ changes with $Y$.

### D.3 FAILURE MODE 1: IDENTICAL HOMOPHILY, DIFFERENT COMPUTATION

Consider two annotated graphs with the same edge set and the same joint distribution of $(X, Z)$. They share identical $h$ and $h_{\text{adj}}$. Now alter only the semantics: in Graph A, $Y$ is independent of $(X, Z)$ and does not affect prediction. in Graph B, $Y$ gates an XOR-like rule so that $X$ is recoverable only from the pair $(Z, Y)$. Homophily cannot distinguish these graphs. A smoothing GNN may succeed on A and fail on B. The profile separates these cases because synergy differs.

### D.4 FAILURE MODE 2: HOMOPHILOUS SELECTION INCREASES HOMOPHILY WHILE COLLAPSING SYNERGY

We analyze the selection mechanism used in the paper. Let $P_{\text{pp}}$ denote the world emission law over $(X, Z, Y)$. Let $E$ be the edge selection event. We observe $P_{\text{ec}}(\cdot) = P_{\text{pp}}(\cdot \mid E = 1)$. Assume the selection probability depends on label agreement:

$$\mathbb{P}(E = 1 \mid X, Z) \propto \exp\left(\beta \, \mathbf{1}\{X = Z\}\right), \qquad \beta \geq 0. \tag{15}$$

Then $h(\beta)$ is non-decreasing in $\beta$ under $P_{\text{ec}}$. Moreover, as $\beta \to \infty$, $P_{\text{ec}}$ concentrates on $X = Z$ whenever the support permits it. Correspondingly in the PID-DCSBM model

$$\mathbb{P}(E_{ij} = 1 \mid X, Z) = \sigma\left(\beta_0 + \log(S_i S_j) + \beta_C \cdot \mathbf{1} X = Z\right) \tag{16}$$

where $\beta_C$ acts as the causal intervention point for homophilous selection.

In that limit,

$$H(X \mid Z) \to 0, \tag{17}$$

and BROJA synergy must vanish because no residual uncertainty remains for a joint rule to resolve. Thus homophily increases as synergy collapses. This is the masking phenomenon. Homophily therefore moves in the opposite direction to the computational difficulty of edge-gated prediction.

Table 2: Synthetic regimes: measured Adjusted Information Profile coordinates (mean±std over 100 graph trials) and edge homophily $h$. UNQ and SYN have indistinguishable $h$ but sharply different AIP coordinates.

| Regime | $\sigma^*$ (Syn) | $\phi^*$ (Red) | $\rho_Y^*$ (Unq) | $h$ |
|--------|------------------|----------------|------------------|-----|
| R2_RED | $0.000 \pm 0.000$ | $0.465 \pm 0.001$ | $0.000 \pm 0.000$ | $0.950 \pm 0.001$ |
| R3_UNQ | $0.028 \pm 0.007$ | $0.008 \pm 0.004$ | $0.571 \pm 0.003$ | $0.250 \pm 0.002$ |
| R4_SYN | $0.987 \pm 0.000$ | $0.002 \pm 0.001$ | $0.000 \pm 0.000$ | $0.250 \pm 0.002$ |

## D.5 FAILURE MODE 3: HOMOPHILY DOES NOT IDENTIFY WHICH INDUCTIVE BIAS IS NEEDED

High homophily can arise in at least two incompatible regimes. First, redundancy-dominant regimes, where many neighbors provide the same evidence for $X$. Second, selection-dominated regimes, where edges exist only when $X = Z$. Both yield large $h$. Only the first favors smoothing as a mechanism. The second can destroy synergy and unique information that would otherwise be available. The profile disambiguates these cases because $\phi^*$ and $\sigma^*$ respond differently.

## D.6 RELATION TO HETEROPHILY BENCHMARKS

The benchmark graphs of Platonov et al. Platonov et al. (2023b) were designed to expose the limits of homophily as a dataset descriptor. Our contribution targets a different axis. We measure edge-conditioned computation, not label mixing. This matters whenever edges encode typed relations or contexts. It also matters when edges are selected through label-dependent mechanisms, as in Topological Masking.

# E EXPERIMENTS

We evaluate whether the *Adjusted Information Profile* (AIP) (i) recovers the intended data-generating regime, (ii) detects *topological masking* under homophilous edge selection, and (iii) predicts which GNN inductive bias is required for node classification. Synthetic experiments use the PID–DCSBM generator (Appendix E); we report averages over 10 graph realizations and 5 random splits per graph (50 runs/model/regime).

## E.1 SYNTHETIC REGIMES: THE PROFILE SEPARATES MECHANISMS THAT HOMOPHILY CANNOT

Table 2 shows the measured AIP coordinates for the three primary synthetic regimes. As designed, **RED** concentrates mass on redundancy ($\phi^*$ high) with strong assortativity ($h \approx 0.95$), while **SYN** concentrates mass on synergy ($\sigma^* \approx 0.987$) despite *near-random* edge homophily ($h \approx 0.25$). Crucially, **UNQ** and **SYN** have essentially the *same* classical homophily ($h \approx 0.25$), yet their AIP footprints are disjoint: UNQ is dominated by unique information ($\rho_Y^* \approx 0.571$) whereas SYN is dominated by synergy ($\sigma^* \approx 0.987$). This is exactly the failure mode for homophily as a computational descriptor: it cannot distinguish "features-only" tasks from truly *relational* (edge-conditioned) tasks.

## E.2 GNN PERFORMANCE: THE SIMPLEX POSITION PREDICTS WHICH INDUCTIVE BIAS IS NEEDED

Table 3 reports node classification accuracy. The results align with standard expectations from the GNN literature *once the regime is identified correctly*: (i) in RED (strong assortativity), smoothing/message-passing models saturate; (ii) in UNQ, topology is a distractor and a simple MLP is near-optimal; (iii) in SYN, plain smoothing (GCN/GAT) is at chance, while an edge-type-conditioned model (RGCN) succeeds.

However, there are two important (and non-cosmetic) implications for how these results should be interpreted: **(a)** In RED, the MLP is also essentially perfect, meaning this regime does *not* isolate

Table 3: GNN accuracy across synthetic regimes (mean±std over 50 runs). Chance is $\approx 25\%$ (4-way classification).

| Regime | GCN | GAT | MLP | RGCN |
|--------|-----|-----|-----|------|
| R2_RED | $100.0 \pm 0.1$ | $100.0 \pm 0.1$ | $99.9 \pm 0.1$ | $98.9 \pm 0.4$ |
| R3_UNQ | $45.3 \pm 16.9$ | $25.7 \pm 3.0$ | $98.7 \pm 0.5$ | $71.9 \pm 2.2$ |
| R4_SYN | $23.4 \pm 1.7$ | $23.4 \pm 1.6$ | $23.8 \pm 1.7$ | $99.9 \pm 0.1$ |

Table 4: **Profiles and model scores on heterophily benchmarks.** Datasets follow Platonov et al. Platonov et al. (2024). We report the Adjusted Information Profile and performance accuracy for smoothing and relational models.

| Dataset | Adj. Information Profile | | | Homophily | Test score | | | | | |
|---------|-------------------------|--|--|-----------|------------|--|--|--|--|--|
| | $\sigma^*$ (Syn) | $\rho_Y^*$ (Unq) | $\phi^*$ (Red) | $h_{\text{adj}}$ | MLP | GCN | GAT | RGCN | SGFormer | RGT |
| **Roman-empire** | 1.000 | 0.287 | 0.000 | -0.05 | 59.7 | 44.2 | 39.9 | 94.7 | 78.5 | **95.1** |
| **Amazon-ratings** | 1.000 | 0.347 | 0.000 | 0.15 | 36.1 | 37.9 | 38.6 | **91.3** | 35.6 | 89.5 |
| **Tolokers** | 0.964 | 0.257 | 0.000 | -0.19 | 71.4 | 69.1 | 78.2 | 99.3 | 70.1 | **99.4** |
| **Minesweeper** | 0.513 | 0.140 | 0.000 | 0.01 | 71.0 | 79.8 | 79.9 | **97.1** | 77.9 | 96.6 |
| **Questions** | 0.297 | 0.078 | 0.000 | -1.77 | 90.8 | 81.1 | 96.9 | **97.9** | 94.3 | 97.2 |

"need for topology"—the label is already recoverable from node features in your current generator settings. **(b)** In UNQ, GCN shows extremely high variance across splits/seeds (std $\approx 0.169$), which is consistent with message passing being an unstable nuisance when the graph is non-informative.

### E.3 Topological masking: homophilous selection suppresses observable synergy

We quantify masking by fixing a highly synergistic *world* mechanism and increasing the homophily selection strength $\beta_C$ (Appendix E). As $\beta_C$ increases, the edge-conditioned law concentrates on $X=Z$, reducing $H(X \mid Z)$ and suppressing the *observable* synergy. Empirically, the observed synergy proxy drops from $\sigma_{\text{ec}} \approx 0.85$ at $\beta_C=0.5$ to $\sigma_{\text{ec}} \approx 0.136$ at $\beta_C=3.0$, while the world synergy stays near 1.0 by construction. (The slight non-monotonicity between $\beta_C=0$ and 0.5 is a finite-sample/estimation artifact and should be presented with CIs.)

### E.4 Heterophily benchmarks: latent lifting

Many benchmark graphs lack explicit edge semantics. To compute profiles, we introduce a lifted edge label $\tilde{Y}_{ij}$ derived from node labels. We evaluate majority-vote lifting and stricter consensus variants. We report majority-vote lifting in the main comparison because it is stable under sparsity. We treat lifting as a measurement instrument, not as ground truth. Appendix D explains why homophily alone cannot substitute for this measurement.

### E.5 Additional results on heterophily benchmarks

Table 4 reports profiles and model performance.

### E.6 Real graphs: PPI and TEG

the PPI graph is derived from the OmniPath Türei et al. (2026) database with nodes prersetnting proteins, and edges interaction between them. Node labels are given by theh amount of a certain protein while the Edge labels are given by the type of interaction ( positive or negative). Various subgraphs were made using the celltype.

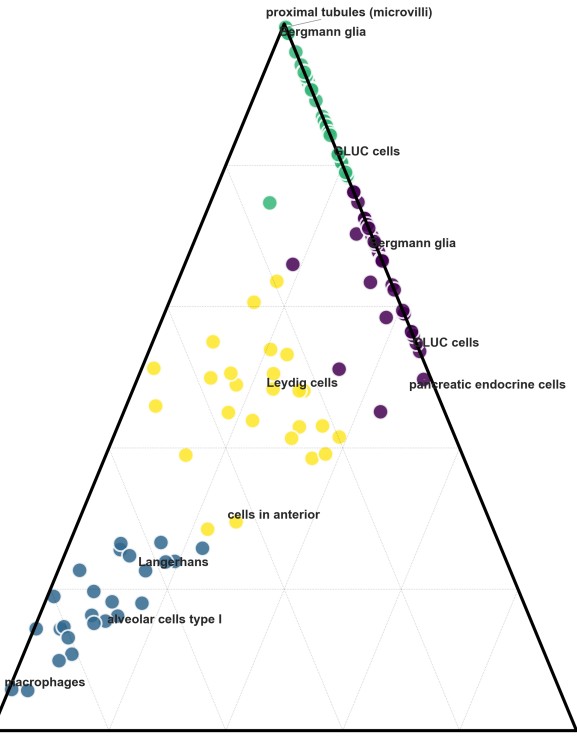

Figure 10: Distinct Information Regimes in Cell-Type Specific Signaling Networks. The Partial Information Decomposition (PID) profile for each cell type network is normalized and mapped onto a 2-simplex. The vertices represent networks dominated by Synergy ($\sigma_*$, Top), corresponding to combinatorial logic; Unique Information ($\rho_*$, Left), corresponding to specific 1-to-1 signaling (homophily); and Redundancy ($\phi_*$, Right), corresponding to shared signals. Points are colored by k-means clusters (k=4), revealing four distinct functional regimes: a Pure Synergy regime (Cluster 2) performing complex computation, a Unique-Dominant regime (Cluster 1) acting as "broadcasters," and two mixed regimes (Clusters 0 & 3).

### E.6.1 PPI

Our analysis of protein-protein interaction (PPI) subgraphs across 146 cell types reveals a functional dichotomy that classical graph metrics fail to capture. While standard label homophily assumes that similarity drives connection, our Partial Information Decomposition (PID) approach demonstrates that structural heterophily often encodes computational *synergy*, whereas homophily correlates with *redundancy*. This trade-off aligns with fundamental biological design principles: systems requiring multi-modal integration maximize synergy, while those prioritizing fail-safe execution maximize redundancy.

**The Inadequacy of Classical Homophily.** Standard homophily metrics prove insufficient for characterizing regulatory logic in biological networks. In our data, high label homophily strongly correlates with redundancy, reflecting duplicative signaling pathways where partner identity alone predicts the outcome. However, homophily fails to detect the sign-gated logic prevalent in complex regulation. We find that the highest synergy values often occur in heterophilic regimes (adjusted homophily < 0). This suggests that "disassortative" mixing in biological graphs is not mere noise, but a structural necessity for integrating diverse, contradictory signals into a coherent cell state.

**Synergistic Integrators ($\Sigma$-dominant).** Cell types functioning as complex logic gates maximize synergy. Bergmann glia and Purkinje cells—the computational cores of the cerebellum—show the

highest synergistic efficiency. Biologically, this reflects the necessity to integrate multimodal inputs (e.g., glutamate, $Ca^{2+}$, pH) to determine plasticity outcomes Nimmerjahn et al. (2009). For Purkinje cells, high synergy captures their role as coincidence detectors, where the firing output depends on the precise conjunction of parallel and climbing fiber signals De Zeeuw et al. (2011).

**Robust Executors ($\Phi$-dominant).** Conversely, systems requiring stability against noise favor redundancy. White matter processes and secretory cells (e.g., Goblet cells) are redundancy-dominant. In white matter, the structural interactome governing myelination and axonal transport utilizes parallel pathways to prevent conduction failure Nave (2010). Similarly, mucin secretion in Goblet cells employs duplicative signaling to guarantee barrier maintenance regardless of single-pathway fluctuations Curran & Cohn (2010).

Table 5: Information Profile of Representative Cell Types. **Integrators** ($\Sigma > \Phi$) manage complex logic; **Executors** ($\Phi > \Sigma$) prioritize robustness. ($\Sigma$: Synergy, $\Phi$: Redundancy).

| Cell Type | Dom. | Ratio ($\Sigma{:}\Phi$) | Biological Interpretation |
|---|---|---|---|
| *Synergistic Integrators (Conditional Logic)* | | | |
| Bergmann glia (mem) | $\Sigma$ | 5.6:1 | Surface signal integration Nimmerjahn et al. (2009) |
| Purkinje cells (nuc) | $\Sigma$ | 5.2:1 | Coincidence detection De Zeeuw et al. (2011) |
| Proximal tubules | $\Sigma$ | >10:1 | Transport regulation |
| *Robust Executors (Fail-safe)* | | | |
| White matter proc. | $\Phi$ | $1{:}\infty$ | Axonal transport stability Nave (2010) |
| Goblet cells | $\Phi$ | $1{:}\infty$ | Mucus barrier maintenance Curran & Cohn (2010) |
| Pancreatic endocrine | $\Phi$ | $1{:}\infty$ | Robust hormone secretion |

### E.6.2 TEXT-ENRICHED GRAPH (TEG)

**Construction.** Citation network from TEG paper Li et al. (2024) We then contruct types scores on the edges using BERT based models, finally derive labels using using openAlex Priem et al. (2022) Nodes: papers with research-area labels (6 classes). Edges: citation relationships annotated with context type via LLM classification: `extends`, `uses`, `compares`, `background`, `contradicts`. Details in Appendix E.

## F GEOMETRIC MOTIVATION

The adjusted profile admits natural interpretations via cellular sheaves and information geometry. This appendix develops these connections; they are not required for the main results but provide deeper structural insight.

### F.1 CELLULAR SHEAVES ON GRAPHS

A *cellular sheaf* $\mathcal{F}$ on graph $G = (V, E)$ assigns a stalk $\mathcal{F}(v)$ to each vertex and a restriction map $\mathcal{F}_{v \triangleleft e} : \mathcal{F}(v) \rightarrow \mathcal{F}(e)$ for each incidence Hansen & Gebhart (2020). In our context: stalks $\mathcal{F}(v)$ represent node label spaces; stalks $\mathcal{F}(e)$ represent edge annotation spaces; restriction maps encode how node labels constrain edge annotations.

### F.2 CONTEXTUAL CAPACITY AS CURVATURE

The contextual capacity $\mathcal{C}_\sigma$ measures how much the annotation channel varies with context $Z$. Geometrically, this is *curvature* of the fibre bundle structure:

- **Flat connection**: $P(Y|X, Z = z) = P(Y|X)$ for all $z$. The annotation channel is context-independent. Analogous to zero curvature in differential geometry.
- **Curved connection**: $P(Y|X, Z = z)$ varies with $z$. Context modulates the channel. Positive curvature.

The flat-bundle null $P_{\text{flat}}$ corresponds to forcing the connection flat. $\mathcal{C}_\sigma - \text{Syn}_{\text{flat}}$ measures the curvature—additional bandwidth from context variation.

**Proposition F.1** (Curvature-capacity correspondence). *Let $\mathcal{R}(P) = \sum_z P(z)D_{\mathrm{KL}}[P(Y|X, Z = z)\|P(Y|X)]$ be the average KL divergence of conditional channels from the marginal channel. Then:*
$$\mathcal{C}_\sigma \leq \mathcal{R}(P) \leq \mathcal{C}_\sigma + H(Y|X).$$
$\mathcal{R}(P) = 0$ *iff the connection is flat.*

### F.3 RESTRICTION MAPS AND MARKOV STRUCTURE

The Markov chain $X \to Z \to Y$ asserts that the restriction map $\mathcal{F}_{X \to Y}$ factors through $\mathcal{F}_{X \to Z}$ and $\mathcal{F}_{Z \to Y}$:
$$\mathcal{F}_{X \to Y} = \mathcal{F}_{Z \to Y} \circ \mathcal{F}_{X \to Z}.$$
When this factorization holds, unique information $\mathrm{Unq}(Y) = 0$. The ratio $\rho_Y^*$ measures failure of this factorization.

**Proposition F.2** (Markov factorization and unique information). $\rho_Y^* = 0$ *iff the stochastic map $P(Y|X, Z)$ factors as $P(Y|Z)$, i.e., the restriction map through $Y$ factors through $Z$.*

### F.4 REDUNDANCY AS CONSISTENT SECTIONS

A *section* of a sheaf assigns values to stalks consistently with restriction maps. Redundancy can be viewed as measuring the space of consistent sections.

**Definition F.1** (Consistent section). A section $s : V \cup E \to \bigcup_c \mathcal{F}(c)$ is consistent if $\mathcal{F}_{v \triangleleft e}(s(v)) = s(e)$ for all incidences.

High redundancy corresponds to large spaces of consistent sections—many ways to assign values that agree across the sheaf structure.

### F.5 SHEAF NEURAL NETWORKS

Standard GNNs implicitly use *constant sheaves*: all stalks are $\mathbb{R}^d$ with identity restriction maps Bodnar et al. (2022). This corresponds to assuming smoothing is appropriate. Our framework diagnoses when this assumption fails:

- High $\phi^*$: Constant sheaf appropriate. Use GCN.
- High $\sigma^*$: Non-trivial sheaf structure. Use sheaf neural networks or edge-conditioned architectures.
- High $\rho_Y^*$: Edge stalks dominate. Use edge-centric models.

**Proposition F.3** (Sheaf Laplacian and smoothing). *The sheaf Laplacian $L_\mathcal{F} = \delta^T \delta$ (where $\delta$ is the coboundary operator) generalizes the graph Laplacian. GCN smoothing minimizes $x^T Lx$; sheaf NNs minimize $x^T L_\mathcal{F} x$. High $\sigma^*$ indicates non-trivial $L_\mathcal{F}$.*

### F.6 INFORMATION GEOMETRY

The capacity normalization connects to information geometry Amari (2016). The space of distributions $P(X, Y, Z)$ forms a statistical manifold with Fisher information metric.

**Proposition F.4** (PID as projection). *PID components correspond to projections onto exponential submanifolds:*

- Syn*: projection onto $\mathcal{M}_{\mathrm{flat}} = \{P : Y \perp Z|X\}$.*

- Unq($Y$)*: projection onto $\mathcal{M}_{\mathrm{Markov}} = \{P : X \to Z \to Y\}$.*

- Red*: projection onto $\mathcal{M}_{\mathrm{ind}} = \{P : Z \perp Y|X\}$.*

The Pythagorean theorem in information geometry Csiszár (1975) implies orthogonality of these projections, providing geometric grounding for the additive PID decomposition.

**Proposition F.5** (Pythagorean decomposition). *For distributions in general position:*
$$D_{\mathrm{KL}}[P\|P_0] = D_{\mathrm{KL}}[P\|P_{\mathrm{flat}}] + D_{\mathrm{KL}}[P_{\mathrm{flat}}\|P_0],$$
*where $P_0$ is the maximum-entropy distribution on $\mathcal{X} \times \mathcal{Z} \times \mathcal{Y}$. This decomposes total "information content" into context-dependent and context-independent components.*

## G  AI usage statement

AI was used to review and assist with the writing/language adn formatting/visuals of the paper.

## H  Data and Code Release

Both the datasets and code will be available on publication. This is a shortened version for the workshop. An extended version will be available on Arxiv.

# I    Science of DL Improvement Challenge Submission

## I.1    What model are you targeting?

*Provide a summary of the problem the deep net model is designed to solve. Good summaries should outline the state of the literature, provide an overview that domain experts would consider reasonable, and cite relevant sources.*

We target **Graph Neural Networks (GNNs)** for node classification on **annotated and relational graphs**. Message Passing Neural Networks (MPNNs), including architectures such as GCN and GAT, dominate this setting and are typically justified through a homophily-based lens: neighborhood aggregation is expected to succeed when adjacent nodes share labels. Within this view, smoothing-based architectures are treated as the default, with architectural modifications introduced primarily to address explicitly heterophilous graphs.

This framing obscures a critical distinction. Homophily metrics reduce graph structure to scalar label agreement and cannot distinguish between label disagreement and *edge-gated computation*. In many scientific domains, including protein interaction networks and citation graphs, edges encode semantics that alter how neighbor information should be interpreted, such as activation versus inhibition or support versus contradiction. In these settings, edges function as computational gates rather than passive conduits.

Our work targets the resulting **architectural mismatch**. Standard diagnostics based on homophily fail to identify when tasks require relational composition, leading to systematic failure modes in which smoothing GNNs are applied to problems that mathematically require edge-conditioned computation. This mismatch explains why architectures such as RGCN or relational Transformers can outperform GCN-style models even when classical homophily statistics appear favorable.

## I.2    How do your results contribute—or could potentially contribute—to understanding these models?

*What aspects of the models become better understood thanks to your work?*

First, we show that failure of smoothing MPNNs is driven by **synergy**, not by low homophily. High-synergy tasks require joint interpretation of neighbor labels and edge annotations.

Second, we identify **Topological Masking**, a causal phenomenon in which homophilous edge selection acts as a collider. This selection process suppresses observable synergy in the edge-conditioned distribution, even when the underlying world mechanism is strongly synergistic. As a consequence, homophily metrics systematically underestimate the need for edge-gated computation.

Third, we introduce the **Information Simplex** as a geometric summary of dataset structure. We show that architectural performance aligns with a dataset's position in this simplex: smoothing models dominate in redundancy-heavy regimes, while relational architectures gain advantage as synergy increases. This provides a principled explanation for recurring empirical patterns, such as the strong performance of relational models on datasets like `Roman-empire`, without relying on post hoc tuning.

## I.3    How do you expect your submission to influence future work?

*Propose ways in which your insights, findings, or methodologies could shape subsequent research directions, model design choices, or scientific applications.*

Our work enforces a paradigm shift from heuristic to measurement-based model design. First, it enables Scientific Architecture Selection, where practitioners compute the profile ex ante to choose between smoothing and edge-gated architectures, yielding accuracy gains from 23% to 99% on synergistic tasks. Second, it necessitates Benchmarking Reform, showing that synthetic "lifting" destroys synergistic information and demanding richly annotated scientific datasets like the MR-PPI and TEG graphs we introduce. Finally, by isolating Unique Edge Information ($\rho_Y^*$), we motivate Next-Generation Architectures like Lifted Graph Transformers that decouple edge and node streams to exploit semantics without topological smoothing.

