# OpenReview forum: "Homophily as a Lossy Channel: Decomposing Information in Graphs and Graph Neural Networks"
_ICLR.cc/2026/Workshop/Sci4DL — Sci4DL 2026_

### Official Review · Reviewer_pvpv · 2026-02-15

**Fit:** 2
**Significance:** 2
**Confidence:** 2

**Summary:**

This paper introduces Adjusted Information Profile (AIP), a Partial Information Decomposition (PID) framework for analyzing how graph structure and edge annotations jointly contribute to node label prediction. By decomposing neighborhood information into redundancy, unique information, and synergy, the AIP allows for comparisons across graphs and exposes the phenomenon of Topological Masking, where homophilous edge selection can suppress observable synergy even when the underlying generative mechanism is highly synergistic. The authors validate their framework through controlled synthetic experiments using both a causal generator and real annotated datasets, demonstrating that the profile is able to predict when simple smoothing GNNs suffice versus when edge-conditioned relational models are necessary.

**Strengths:**

- **Relevant theoretical framework**: The paper provides a well-supported theoretical framework that generalizes homophily-based analyses using principles from information theory.
- **Causal PID–DCSBM generator**: The authors developed a new synthetic graph generator that allows for "intervening" on specific information types. This provides a "ground truth" to prove that their metrics actually detect what they claim to detect.
- **Good experimental design and presentation of the results**: The experimental design is thorough, with comparisons between smoothing (GCN/GAT) and edge-conditioned relational (RGCN/transformer) architectures. Authors propose three different claims that are well supported by the results.

**Suggestions:**

- **Problem setup and motivation**: The problem setup and motivation could be made clearer. While contributions are listed explicitly, the introduction sometimes jumps between ideas, making it challenging to contextualize the work for readers more unfamiliar with PID or information-theoretic analyses in graphs. ﻿The introduction and background sometimes jump between concepts (homophily, PID, edge annotations, causal models) a bit abruptly, which slightly difficults the reading flow. Furthermore, Figures 1 and 2 are not mentioned or contextualized in the text either.
- **Practical relevance**: The practical relevance of the Adjusted Information Profile could be more explicitly connected to real-world use cases. Authors could better discuss how their proposed approach could be leveraged in the future (e.g would it be useful for GNN design, model selection, graph analysis…?)
- **Incomplete data**: As follow-up work, it would be interesting to explore the robustness of the metrics under noisy or incomplete data, to show how sensitive the AIP is to real-world graph imperfections.
- (Very minor) Some cross-references in the appendix are broken

---

### Official Review · Reviewer_aib8 · 2026-02-20

**Fit:** 2
**Significance:** 2
**Confidence:** 1

**Summary:**

This paper argues that standard homophily metrics are insufficient descriptors of graph structure for GNN architecture selection, because they only capture label agreement between neighboring nodes and completely ignore edge annotations. The authors propose the Adjusted Information Profile (AIP), and show that AIP predicts when smoothing architectures (GCN) suffice versus when edge-conditioned models (RGCN) are required. They also identify and formalize Topological Masking: homophilous edge selection acts as a causal collider that suppresses observable synergy even when the underlying mechanism is highly synergistic. The framework is validated on synthetic graphs with controlled information structure, heterophily benchmarks, and two new annotated datasets (a text-enriched citation graph and a protein interaction network).

**Strengths:**

- The core observation is well-motivated and practically relevant: homophily ignores edge annotations entirely, and this omission is consequential whenever edges carry semantic meaning that gates how neighbor information should be interpreted.
- The Topological Masking phenomenon is interesting, as it shows that homophily and synergy can move in opposite directions, meaning homophily is not just uninformative but actively misleading as a proxy for computational difficulty.
- The theoretical grounding is solid, with formal proofs of boundedness, consistency, and calibration under null models in the appendix.

**Suggestions:**

- The AIP is defined over the triple $(X, Z, Y)$, which requires edge annotations to be available. For the heterophily benchmarks, where annotations are absent, the authors resort to "latent lifting", i.e., inferring a coarse edge label from node labels. This implies that the lifted labels are derived from the node labels, which renders a circular issue. This is a significant limitation of the analysis and proposed methodology for real-world graphs.
- The AIP involves several non-trivial computational steps: BROJA PID (a convex optimization), Blahut-Arimoto for capacity estimation, and discretization of continuous variables. The sensitivity of the profile to discretization choices (number of bins, smoothing parameter) is not thoroughly studied, and it is unclear how robust the results are to these choices in practice.
- The paper is dense and the notation is heavy, which makes it difficult to follow, especially in the main body where several key quantities are defined compactly without sufficient intuition. An intuitive explanation alongside a toy example throughout the paper would help to grasp better each concept, beyond the description in the appendix (or at least add a reference to the corresponding section in the Appendix).
- Several appendix references are broken (`Appendix ??`).

---

### Meta-Review · Area_Chair_uLhD · 2026-03-01

**Recommendation:** Accept

**Metareview:**

Recommending accept, due to it being a decent fit for the workshop.

---

### Decision · Program_Chairs · 2026-03-02

Accept